

# Multi-instrument comparison and compilation of non-methane organic gas emissions from biomass burning and implications for smoke-derived secondary organic aerosol precursors

Lindsay E. Hatch[1*], Robert J. Yokelson[2], Chelsea E. Stockwell[2§], Patrick R. Veres[3,4], Isobel J. Simpson[5], Donald R. Blake[5], John J. Orlando[6], Kelley C. Barsanti[1*]

[1]Department of Civil and Environmental Engineering, Portland State University, Portland, OR, USA
[2]Department of Chemistry, University of Montana, Missoula, MT, USA
[3]Cooperative Institute for Research in Environmental Sciences, University of Colorado, Boulder, CO, USA
[4]Chemical Sciences Division, Earth System Research Laboratory, National Oceanic and Atmospheric Administration, Boulder, CO, USA
[5]Department of Chemistry, University of California-Irvine, Irvine, CA, USA
[6]National Center for Atmospheric Research, Boulder, CO, USA
[*]Current address: Department of Chemical and Environmental Engineering and College of Engineering – Center for Environmental Research and Technology (CE-CERT), University of California- Riverside, Riverside, CA, USA
[§]Current address: Chemical Sciences Division, Earth System Research Laboratory, National Oceanic and Atmospheric Administration, Boulder, CO, USA

*Correspondence to*: Kelley C. Barsanti (kbarsanti@engr.ucr.edu)

**Abstract.** Multiple trace-gas instruments were deployed during the fourth Fire Lab at Missoula Experiment (FLAME-4), including the first application of proton-transfer-reaction time-of-flight mass spectrometry (PTR-TOFMS) and comprehensive two-dimensional gas chromatography-time-of-flight mass spectrometry (GC×GC-TOFMS) for laboratory biomass burning (BB) measurements. Open-path Fourier-transform infrared spectroscopy (OP-FTIR) was also deployed, as well as whole air sampling (WAS) with one-dimensional gas chromatography-mass spectrometry (GC-MS) analysis. This combination of instruments provided an unprecedented level of detection and chemical speciation. The chemical composition and emission factors (EFs) determined by these four analytical techniques were compared for four representative fuels. The results demonstrate that the instruments are highly complementary, with each covering some unique and important ranges of compositional space, thus demonstrating the need for multi-instrument approaches to adequately characterize BB smoke emissions. Emission factors for overlapping compounds generally compared within experimental uncertainty, despite some outliers, including monoterpenes.

Data from all measurements were synthesized into a single EF database that includes over 500 non-methane organic gases (NMOGs) to provide a comprehensive picture of speciated, gaseous BB emissions. The identified compounds were assessed as a function of volatility; 6-11% of the total NMOG EF was associated with intermediate volatility organic compounds (IVOCs). These atmospherically relevant compounds historically have been unresolved in BB smoke measurements and thus are largely missing from emissions inventories. Additionally, the identified compounds were screened for published secondary organic aerosol (SOA) yields. Of the total reactive carbon (defined as EF scaled by the OH rate constant and



carbon number of each compound) in the BB emissions, 55-77% was associated with compounds for which SOA yields are unknown or understudied. The best candidates for future smog chamber experiments were identified based on the relative abundance and ubiquity of the understudied compounds, and included furfural, 2-methyl furan, 2-furan methanol, and 1,3-cyclopentadiene. Laboratory study of these compounds will facilitate future modeling efforts.

**1 Introduction**

Biomass burning (BB) emits large amounts of trace gases, including non-methane organic gases (NMOGs) and primary (directly emitted) particulate matter (PM). NMOGs also react in the atmosphere to form secondary PM and ozone. BB-PM has been difficult to represent accurately in models used for chemistry and climate predictions (Alvarado et al., 2009; Alvarado et al., 2015; Heald et al., 2011; Reddington et al., 2016), including for air quality and fire management purposes.

Given the significant influence of PM on the radiative balance of the atmosphere (Hobbs et al., 2003) and on cloud formation (Desalmand and Serpolay, 1985; Reid et al., 2005), as well as on human health (Naeher et al., 2007; Tinling et al., 2016; Viswanathan et al., 2006), more accurate model representation of BB-PM is needed. This is particularly true given the projected increase in fire activity globally due to increased food demand (Tilman et al., 2001) and climate change (Flannigan et al., 2009; Hessl, 2011; Westerling et al., 2006; Yue et al., 2015).

While many factors contribute to the challenge of accurately predicting BB-PM (Herron-Thorpe et al., 2014), one significant limitation has been the incomplete identification and quantification of NMOGs emitted from fires that may serve as precursors for secondary organic PM (i.e., secondary organic aerosol, SOA) (Alvarado and Prinn, 2009; Alvarado et al., 2009; Warneke et al., 2011). Given that BB is the second largest source of NMOGs worldwide, the SOA formation potential from BB is large (Yokelson et al., 2008), yet poorly understood. Much recent research supports that many previously

unconsidered SOA precursors exist (Chan et al., 2009; Lim and Ziemann, 2009; Robinson et al., 2007), and that mechanisms beyond gas/particle partitioning of semi-volatile organic compounds contribute to ambient SOA formation, including oxidation of lower volatility precursors (Ziemann and Atkinson, 2012; Robinson et al., 2007). More specifically, it has been demonstrated that the unspeciated NMOGs may contribute significantly to BB-SOA (Jathar et al., 2014). In order to accurately model the production of BB-SOA, as well as other secondary pollutants (e.g., ozone and peroxyacyl nitrates),

improved identification and quantification (e.g., emission factors, EFs) are needed for all compounds/classes of compounds that can serve as SOA precursors.

In this work, the determination of previously un- and under-characterized gas-phase organic compounds and compound classes was pursued by extensive analysis and synthesis of data collected from a unique and powerful combination of techniques. This work builds on prior BB emissions characterization efforts (e.g., (Yokelson et al., 2013) in which high

molecular weight NMOGs were detected but many (30-70% by mass) could not be identified. NMOGs emitted from laboratory biomass burns were measured during the fourth Fire Lab at Missoula Experiment (FLAME-4) using: open-path Fourier-transform infrared spectroscopy (OP-FTIR) (Stockwell et al., 2014), whole air sampling with 1-D GC analysis



(WAS), proton-transfer-reaction time-of-flight mass spectrometry (PTR-TOFMS) (Stockwell et al., 2015), and comprehensive two-dimensional gas chromatography/time-of-flight mass spectrometry (GC×GC-TOFMS) (Hatch et al., 2015). The data were analyzed and synthesized herein to meet the following objectives: (1) compare the compositional space and calculated EFs accessed by each instrument; (2) provide comprehensive BB gas-phase emissions profiles for each

of the sampled fuels; and (3) describe the volatility distribution of the determined compounds and identify potentially important, yet understudied SOA precursors.

## 2 Methods

### 2.1 FLAME-4 Sampling

During FLAME-4, two burning configurations were utilized: stack burns and room burns (Stockwell et al., 2014). Data

included here were obtained during room burns wherein smoke from flaming and smoldering combustion mixed throughout the burn chamber. The smoke was "stored" in the room for approximately two hours while sampling occurred; thus some lower volatility compounds were eventually lost to particles or surfaces (Stockwell et al., 2014). Four burns were chosen for in-depth analysis: ponderosa pine (*pinus ponderosa*, burn 144, hereafter referred to as pine), Chinese rice straw (*oryza sativa*, burn 153, straw), Indonesian peat (burn 154, peat), and black spruce (*picea mariana*, burn 155, spruce). These

selected fires burned the most globally relevant fuels out of the limited number of burns where gas-phase data were available from all of the above instruments.

Although all instruments sampled during each burn, the timing and location of each sample varied due to the sampling configuration and duration of room burns. An example of the relative sampling periods is provided in Fig. S1. The OP-FTIR measured continuously throughout the burn and was located on a platform high up in the combustion chamber. For

GC×GC-TOFMS, integrated samples were collected closer to the fuel source after mixing was achieved. The PTR-TOFMS sampled spatially near the GC×GC-TOFMS, but often not temporally as it also sampled from two smog chambers throughout the sampling period. WAS canister samples were collected from the smog chambers, which were filled with well-mixed smoke. Although the FLAME-4 measurements were not set up for a rigorous intercomparison, and thus spatial and temporal overlap between the various techniques was not ideal, an assessment of the general agreement and the

compositional space probed by each technique provides new and valuable insights.

### 2.1.1 OP-FTIR

The OP-FTIR system deployed in FLAME-4 was described by (Stockwell et al., 2014). Briefly, it consisted of a Bruker Matrix-M infrared (IR) cube spectrometer with an open White cell that was positioned in a well-mixed part of the combustion room about 15 m above the fuel bed and 10 m from the other instrument inlets. The optical path length was 58.0

m and IR spectra were collected at a resolution of 0.67 cm$^{-1}$. Sixteen interferograms were co-added to give single-digit ppbv detection limits at a time resolution of 6 s with a duty cycle greater than 95%.





Mixing ratios were determined for 19 gas-phase species (and water) by multicomponent fits to selected regions of the IR transmission spectra with synthetic calibration using a nonlinear least squares method (Burling et al., 2010; Stockwell et al., 2014). The uncertainties in the individual mixing ratios vary by spectrum and molecule and are dominated by uncertainty in the reference spectra (1-5%) or the detection limit (0.5-15 ppb), whichever is larger.

OP-FTIR offers numerous advantages for the analysis of BB emissions (Burling et al., 2010; Christian et al., 2004). This approach achieves simultaneous and quantitative measurement of reactive and stable species (both inorganic and organic in nature) from flaming and smoldering combustion with high time resolution. Each analyte's IR spectrum displays multiple unique features, which limits spectral interference when combined with advanced, multi-component chemometric analysis. Further, because of the open-path configuration, OP-FTIR measurements are not subject to storage or sampling losses.

However, foregoing pre-concentration to preserve detection of reactive species limits quantification to compounds present at mixing ratios of several ppb or greater.

### 2.1.2 WAS

During FLAME-4, WAS samples were collected from smog chambers. The smog chambers were filled using Dekati ejector dilutors (Hennigan et al., 2011; Stockwell et al., 2014) situated within the combustion chamber; the smoke was diluted ~25-

fold. The WAS samples were collected into evacuated 2 L electropolished stainless steel canisters and analyzed at the University of California, Irvine (UCI) using multi-column gas chromatography (GC) to measure $CO_2$, CO, $CH_4$, and approximately 70 NMOGs. Details of canister preparation for field and analytical procedures are given in (Simpson et al., 2010). A background canister sample was taken prior to filling the smog chamber and the sample of primary BB emissions was taken immediately before initiation of the chemical perturbation. Carbon dioxide, $CH_4$ and CO were analyzed

separately from the NMOGs using GC with a thermal conductivity detector (TCD) for $CO_2$, and GC with flame ionization detection (FID) for CO and $CH_4$. NMOGs were analyzed by cryogenically preconcentrating 217 $cm^3$ of sample air, then vaporizing the sample with a hot water bath and splitting the air into five different streams, each directed to a different column/detector combination. These include two GC/FID combinations, two GC with electron capture detector (ECD) combinations, and GC with mass spectrometer detection (MSD). The measurement precision, accuracy and detection limits

vary by compound. The detection limit is 3 pptv for NMOGs. The accuracy is 2% for $CO_2$, 1% for $CH_4$, and 5% for CO and NMOGs. The measurement precision is 2% for $CO_2$ and CO, 0.1% for $CH_4$, 3% for most NMOGs (Simpson et al., 2014).

The UCI WAS collection and analysis methods have been rigorously characterized and validated (Simpson et al., 2010). The multi-column and multi-detector approach provides accurate identification and quantification for a range of speciated hydrocarbons and some oxygenates in BB emissions (Simpson et al., 2011). In this work, organonitrates were the only

oxygenates quantified; other oxygenates, such as methanol and acetone, were not quantified because of their higher measurement uncertainty (Simpson et al., 2011) especially for concentrated samples. The "grab" sampling approach limits the temporal coverage of a smoke plume, unless a large number of samples are collected. However, in this work the sampled





smoke was well mixed and therefore a single grab sample is expected to be representative of the overall emissions from all burn phases.

### 2.1.3 PTR-TOFMS

PTR-TOFMS sampling during FLAME-4 has been described in detail (Stockwell et al., 2015). Briefly, a PTR-TOFMS
8000 (Ionicon Analytik GmbH, Innsbruck, Austria) (Jordan et al., 2009) sampled continuously through 1 m, heated (80 °C) PEEK tubing from the control room along one side of the combustion chamber. During the room burns discussed in this work, the PTR-TOFMS sampled intermittently between two smog chambers and the combustion chamber. The mass resolution ($m/\Delta m$) was 4000 - 5000 at $m/z$ 21, with a typical mass range from $m/z$ 10 to 600. The drift tube was operated at 600 V, 2.3 mbar, and 80 °C ($E/N \sim$ 136 Td; $E$ is the electric field strength, $N$ is the concentration of neutral gas, and 1 Td =
$10^{-17}$ V cm$^2$).

The PTR-TOFMS was calibrated every few days using a mixture of formaldehyde (HCHO), methanol ($CH_3OH$), acetonitrile ($CH_3CN$), acetaldehyde ($CH_3CHO$), acetone ($C_3H_6O$), dimethyl sulfide (DMS, $C_2H_6S$), isoprene ($C_5H_8$), methyl vinyl ketone ($C_4H_6O$), methyl ethyl ketone ($C_4H_8O$), benzene ($C_6H_6$), toluene ($C_6H_5CH_3$), $p$-xylene ($C_8H_{10}$), 1,3,5- trimethylbenzene ($C_9H_{12}$), and α-pinene ($C_{10}H_{16}$). Separate mass-dependent calibration factors were derived for hydrocarbons and compounds
that included heteroatoms to calibrate the remaining species; measurement error was estimated to be ~20-30% for calibrated gases and up to 50% for uncalibrated gases (Stockwell et al., 2015). Instrument zeros were periodically performed using a precious metal catalyst.

Proton-transfer-reaction mass spectrometry uses $H_3O^+$-based ion-molecule reactions to ionize analyte species with minimal fragmentation. Only compounds with proton affinities greater than that of water are ionized (de Gouw and Warneke, 2007;
Lindinger et al., 1998). The high mass resolution of the TOF mass analyzer permits separation of compounds that are isobaric at unit mass resolution and enables assignment of molecular formulas, although this method is unable to separate isomers with the same chemical formula.

### 2.1.4 GC×GC-TOFMS

NMOG samples were collected onto adsorption/thermal desorption (ATD) cartridges, as described in (Hatch et al., 2015).
Briefly, cartridge samples were collected from the control room, through a Teflon inlet < 5 m long with the sampling tip located ~2-3 m from the PTR-TOFMS inlet and about 1 m into the burn chamber. To prevent particles and ozone from reaching the sorbent, a glass-fiber filter coated with sodium thiosulfate was placed upstream of the cartridge in the sampling train (Helmig, 1997). The samples were frozen and then analyzed at Portland State University within 1 month of sampling. An ATD 400 system (Perkin-Elmer, Waltham, MA) was used to desorb and inject each sample into a Pegasus 4D GC×GC-
TOFMS (Leco Corp., St. Joseph, MI). Calibration curves were determined for ~275 standard compounds; tentatively identified compounds were calibrated using surrogate standards. Minimum errors of 20% and 50% were assigned for



calibrated and tentatively identified compounds, respectively. The analytical conditions for the pine, straw, and peat smoke samples followed those described by (Hatch et al., 2015); analysis of the spruce smoke sample included here was slightly different and is described in the Supplementary Information.

Key advantages of GC×GC–TOFMS include improved chromatographic separation and sensitivity compared to 1-D GC,
deconvolution capability provided by the high TOFMS spectral collection rate, and the formation of patterns of like compounds in the 2-D retention space that aid in compound classification (Mondello et al., 2008). Therefore, this technique is ideal for speciation of the large number of compounds and isomers emitted from BB (Hatch et al., 2015). However, important polar compounds may adsorb to the glass-fiber filter or may not elute from the GC columns and light compounds may "breakthrough" the sorbent bed, limiting the range of compounds that can be detected (Hatch et al., 2015). Further,
collection of NMOGs onto cartridges yields samples integrated over several minutes or longer, which hinders the ability to capture rapid changes in smoke concentration. However, rapid changes were not expected during the room burn experiments sampled in this work once the smoke was well mixed.

**2.2 Emission Factor Calculations**

Emission factors (EFs) were calculated by the carbon mass balance method (CMB), as described for the OP-FTIR
(Stockwell et al., 2014), PTR-TOFMS (Stockwell et al., 2015), and GC×GC-TOFMS (Hatch et al., 2015) measurements. EFs for the WAS measurements of the spruce smoke sample were also calculated by CMB (Eq. 1):

$$EF_X = F_C \times \frac{MW_X}{MW_C} \times \frac{\frac{\Delta X}{\Delta CO}}{\sum_i^n \left( CN_i \times \frac{\Delta Y_i}{\Delta CO} \right)}. \tag{1}$$

$F_C$ is the mass fraction (g/kg) of carbon in the dry fuel and was measured for each fuel by an independent laboratory. $MW_X$ and $MW_C$ are the molecular weights of compound $X$ and carbon, respectively. $\Delta X$ is the background-subtracted ("excess")
mixing ratio of compound $X$; $\Delta X/\Delta CO$ (or $\Delta Y/\Delta CO$) is the emission ratio (ER) of compound $X$ (or $Y$) relative to CO. $CN_i$ is the carbon number in compound $Y_i$. The summation represents the total carbon emitted during combustion, assuming complete volatilization. Because the WAS sampling methods are capable of measuring $CO_2$, CO, methane, and light hydrocarbons, all data necessary for CMB is generally included in the WAS measurements (Simpson et al., 2011). However, due to smoke dilution upon filling the smog chambers, the WAS $CO_2$ and methane measurements were below or
similar to background levels for the pine, peat, and straw smoke samples. The OP-FTIR-measured $CO_2$ and $CH_4$ concentrations could not be substituted directly because of the different dilution ratios between the combustion chamber (OP-FTIR) and smog chamber (WAS) and therefore CMB was not applied to the WAS dataset for these three burns. WAS CO measurements were always well above the corresponding background concentrations. Thus for the pine, peat, and straw burns, WAS EFs were calculated via CO-based emission ratios and the OP-FTIR CO EF ($EF_{CO}$), as:

$$EF_x = \frac{MW_x}{MW_{CO}} \times \frac{\Delta X}{\Delta CO} \times EF_{CO}. \tag{2}$$





### 2.3 Data Combination and Reduction

Although data and calculated EFs from three of the instruments are available individually (Hatch et al., 2015; Stockwell et al., 2015; Stockwell et al., 2014), merging into a single, combined BB emissions database will allow a more complete representation of BB emissions and subsequent atmospheric chemistry. To that end, overlapping measurements of the same

species must be counted only once to the best possible extent. Data reduction largely followed the approach described by (Yokelson et al., 2013). Because of the open-path configuration, the OP-FTIR is not subject to sampling line artifacts. It is also the only instrument that sampled in real time for the duration of each burn (Fig. S1). Therefore all OP-FTIR data were given precedence, and EFs determined from the other measurements were discarded for the overlapping compounds due to the greater potential for sampling artifacts. To combine the PTR-TOFMS measurements with speciated data from the GC

techniques, the EFs were compared at each chemical formula, summed over all corresponding isomers measured by the GC×GC-TOFMS and/or WAS instruments. If the PTR-TOFMS EF was more than 2× the integrated GC×GC-TOFMS or WAS EF, both measurements were retained, unless a negative artifact was known to affect the GC data (e.g., cartridge breakthrough), in which case only the PTR-TOFMS measurement was used in the combined EF database. This approach preserves speciated information while retaining the potential for additional unknown emissions unaccounted for by the GC

techniques. It is possible such cases may reflect an incorrect calibration (or sampling artifact) in one or both instruments and thus compounds may be double counted in some of these cases. For cases in which the PTR-TOFMS EF was less than 2× that of the GC×GC-TOFMS or WAS EF, the GC data were used to preserve isomer speciation and the PTR-TOFMS measurement was deleted from the synthesized EF database. However, if only one (predominant) isomer was observed in the GC dataset (e.g., $C_6H_6$, benzene), the higher EF was used. For isomer groups detected by both GC×GC-TOFMS and

WAS, the GC×GC-TOFMS EFs were retained if many more isomers were observed by this technique; if the number of observed isomers was similar at a given molecular formula, the measurement yielding the higher total EF was used in the EF database. This filtering approach for building a combined database incurs some error, but the errors tend to cancel (Yokelson et al., 2013).

### 3 Results & Discussion

**3.1 Historical Assessment of BB Emissions Measurements**

In a survey of all publications reporting BB NMOG emissions, species at only a limited number of masses are commonly reported. The compilation (Fig. 1a), which includes 62 publications dating back to year 2000 (not including review articles), represents the percentage of those publications reporting a quantified NMOG (i.e., concentration, mixing ratio, emission ratio, or emission factor) at the indicated mass. Compounds were lumped by nominal mass; thus multiple compounds can

contribute to each molecular weight bin, although each publication is counted only once per bin if more than one isobaric compound was reported. Despite the fact that recent mass spectra of smoke have shown multiple peaks at virtually every





mass (Stockwell et al., 2015; Yokelson et al., 2013), only nine masses are included in over 50% of the publications; 21 masses are reported over 30% of the time. The compounds at these 21 commonly reported masses are all of relatively low molecular weight: only two of them are >100 g/mol.

To demonstrate the volatility range of commonly measured species, we use the compounds compiled in Table 1 of (Akagi et
al., 2011) as a generous representation of typically reported compounds. The saturation concentration ($C^*$) of each compound was estimated using the parameterization described by (Li et al., 2016), which is based solely on molecular formulas and thus can be readily applied to both identified and unidentified compounds. In this approach, compounds with the same number of carbon, nitrogen, and oxygen atoms will be assigned the same $C^*$ value, regardless of chemical structure or degree of unsaturation. Because halogen atoms are not included in this volatility parameterization, halogenated
compounds have been omitted from this assessment. Compounds are plotted in molecular corridors as a function of molecular weight (MW) (Li et al., 2016; Shiraiwa et al., 2014) (Fig. 1b). The dashed lines reflect the parameterized change in $C^*$ for compounds with O:C = 0 (blue) and O:C = 1 (red) with respect to MW (Shiraiwa et al., 2014). Regions of $C^*$-MW space associated with volatile organic compounds (VOCs), intermediate volatility compounds (IVOCs), and semi-volatile organic compounds (SVOCs) are shaded for reference (based on the volatility classifications in Donahue et al. (2009). As
seen in Fig. 1b, nearly all of the routinely measured species can be classified as VOCs. The five compounds within the IVOC range are organonitrates and are likely misclassified as IVOCs using this parameterization. For example, the parameterized $\log_{10}C^*$ value of methyl nitrate is 5.05 compared to 8.95 based on the predicted vapor pressure from ChemSpider (http://www.chemspider.com/Chemical-Structure.11231.html). Fig. 1 illustrates that traditionally applied measurement approaches miss intermediate to semi-volatile organic compounds, including SOA precursors, which are
probed using the combined instrumental analysis described in this work (and plotted in Fig. 1c).

**3.2 Instrument Comparison: Scope and Overlapping Species**

**3.2.1 Overall Comparison**

Figure 2a shows the range of compounds measured by each instrument, as a function of carbon number (CN) and H:C ratio, as well as O:C ratio (marker size). Taken together, the instruments yield data for $CO_2$, CO, $CH_4$, and NMOGs from $C_1$-$C_{15}$,
including compounds with a wide range of double bond equivalents (DBE, 0-7) and O:C ratios (0-3; methyl nitrate contributes the highest O:C ratio) (Fig. 2a). Further, each instrument detected unique compounds and/or covered unique regions in CN-H:C space. The WAS technique measured organonitrates (large triangles in Fig. 2a), as well as light hydrocarbons (HCs), particularly alkanes $\leq C_4$. GC×GC-TOFMS measured the highest MW HCs, including alkanes, alkenes, and sesquiterpenes, whereas the PTR-TOFMS measured more polar compounds. In this study, the OP-FTIR contributed the
data needed for CMB EF calculations for the PTR-TOFMS and GC×GC-TOFMS (i.e., CO, $CO_2$, and $CH_4$), as well as light oxygenates, such as formic and acetic acids, glycolaldehyde, and formaldehyde.





The coverage of each instrument as a function of compound volatility is also shown in Table 1 where the values represent the percentage of the total EF measured by a given instrument relative to the total EF determined from the combined dataset following data synthesis. For this representation, percentages include EFs for overlapping species as detected by each instrument, even if they were eliminated from the combined database during data reduction. Values in parentheses include

EFs determined by OP-FTIR for overlapping compounds that the indicated instrument is capable of measuring, but that were not quantified in this study (i.e., overlapping compounds between PTR-TOFMS and OP-FTIR (Stockwell et al., 2015) and $CO_2 + CH_4$ in the WAS data for reasons discussed in Section 2.2). 'All Compounds' represent the sum of NMOGs, CO, $CO_2$, and $CH_4$. The 'All Compounds' category is dominated by CO, $CO_2$, and $CH_4$ (see also Figs. 4 and 5), which typically constitute >97% of the total carbon emitted by BB (Akagi et al., 2011; Yokelson et al., 2013; Gilman et al., 2015). The OP-

FTIR and WAS samples (with $CO_2$ and $CH_4$ included) detected ~98-99% of the total gas-phase EF. For the NMOG and IVOC categories, PTR-TOFMS generally measured the highest fraction of the total EF regardless of whether the OP-FTIR overlapping species were included or not (Table 1). The peat burn was the only case for which the GC measurements accounted for a similar fraction of the total NMOG EF, due to the higher contribution of alkanes than for the other smoke samples (32% of the total NMOG EF compared to <6% for the other fuels). Because PTRMS instruments using $H_3O^+$

reagent ions are not sensitive to alkanes (Arnold et al., 1998), this major class of compounds would be entirely unaccounted for if only PTRMS measurements are used to measure peat smoke.

In addition to mass closure, speciation of the observed compounds is required for understanding chemical reaction pathways. Figure 2b shows the number of isomers and the contribution of the top isomer to the total EF at that molecular formula, as determined by the chromatographic methods for each molecular formula that overlapped with PTR-TOFMS. Note that there

are some polar compounds for which GC×GC-TOFMS likely missed a dominant isomer (e.g., catechol at $C_6H_6O_2$), which would bias this analysis for a few compounds. To illustrate the relative abundance of each isomer group, the marker sizes in Fig. 2b are proportional to the percent contribution of each group (based on the GC EFs) to the total NMOG EF from Table 1.

For 33% (peat) – 46% (pine) of the 56-60 *m/z* ratios per fuel included in the comparison, 4+ (and up to 32) isomers could be

observed chromatographically. In contrast, only 22% (straw) – 38% (peat) of all included *m/z* ratios corresponded to a single isomer in the GC datasets, although some of the most abundant isomer groups can be reasonably treated as a single isomer despite the presence of multiple minor isomers (top left corner of Fig. 2b; e.g., $C_6H_6$ – benzene and $C_7H_8$ – toluene). However, many relatively abundant isomer groups were not dominated by a single isomer. Particularly in the 4-10 isomer range, many isomer groups that represent a significant portion of the NMOG EF were observed wherein the top isomer

contributed only ~25-75% of the EF for that group (Fig 2b). For groups with 10+ observed isomers, which were overwhelmingly hydrocarbons, the range decreased to only ~15-60% (Fig. 2b), although such groups represent a relatively small percentage of the NMOG EF, with the notable exception of the monoterpenes (Fig. 2b). In spruce and pine smoke, monoterpenes made the largest contribution to the total EF (4.8% and 3.1%, respectively, based on the GC×GC-TOFMS EFs) and had the highest number of isomers among the compounds included in Fig. 2b, with the top isomer contributing





<30% of the total monoterpene EF. Therefore, a number of important isomers were detected chromatographically for many of the overlapping *m/z* ratios observed by PTR-TOFMS, highlighting the difficulty in determining specific compounds using chemical ionization. Future studies that includes a larger number of sampled fires could probe the variability of the isomer distribution within each isomer group to determine the conditions/fuels for which scaling factors could be reasonably applied

in order to coarsely speciate PTR-TOFMS data.

### 3.2.2 Instrument vs. Instrument

Figure 3a shows the correlation between the EFs calculated based on the GCs and OP-FTIR/PTR-TOFMS data; statistics of the comparison for each instrument pair are provided in Table 2. Each PTR-TOFMS EF is compared to the sum of EFs of all isomers at the same molecular formula, as measured by the respective GC instruments. Including overlapping

compounds among all four instruments, 65-72 unique molecular formulas are included in the comparison for each fuel, making this the most comprehensive comparison of BB emissions to date and the first to include data from these specific analytical approaches.

Significant overlap with the OP-FTIR measurements is only available for the WAS dataset (Table 2). These two techniques are the most established and well characterized of the four, and displayed the best correlation among all instrument pairs

(slope = 1.01 ± 0.001, $r^2$ = 1.0, Table 2), despite measuring smoke at different dilution ratios. Only furan overlaps between the GC×GC-TOFMS and OP-FTIR; thus the correlation between these instruments was not assessed. Because PTR-TOFMS-derived EFs were not calculated for the few compounds that overlap with the OP-FTIR (Stockwell et al., 2015), comparison of these two instruments is not available; however (Stockwell et al., 2015) previously reported a strong correlation between the OP-FTIR and PTR-TOFMS methanol data during the FLAME-4 stack burns.

The correlation between the GC×GC-TOFMS and WAS data is given in Fig. S2 and demonstrates good agreement between these two methods for overlapping isomers (slope = 1.32 ± 0.08, $r^2$ = 0.82, Table 2). All data points with the largest discrepancy occurred during the peat burn (Fig. S2). When the peat smoke data points are removed from the linear regression, the slope and $R^2$ improve to 1.11 and 0.95, respectively, indicating that these techniques generally agreed within ~10% among the overlapping isomers (i.e., within the reported uncertainty for the GC×GC-TOFMS data). The reason for

the larger discrepancy in the peat smoke measurements is not entirely clear. Given the multi-column and multi-detector analysis of the canister samples (see section 2.1.2), the likelihood of interferences in the WAS detection is significantly reduced. However, because the peat burn produced the lowest smoke concentrations, WAS-measured excess mixing ratios were significantly lower than for the other burns and thus potentially subject to greater uncertainty given the additional dilution upon filling the smog chambers. Further, because the WAS canister samples were collected from the smog

chambers, rather than directly from the combustion chamber, we cannot rule out the potential that the analyte concentrations were different than those measured by GC×GC-TOFMS during the peat burn (after accounting for dilution; e.g., due to contamination during the chamber fill), although the other burns did not appear to be impacted based on the good agreement between the two methods (Fig. S2). It is also possible that poor isomer separation, poor mass spectral deconvolution, or




incorrect isomer assignments impacted the GC×GC-TOFMS calibration. Future experiments should compare these techniques side-by-side.

The 12 overlapping compounds between WAS and PTR-TOFMS included hydrocarbons and dimethyl sulfide (DMS). The relatively low $R^2$ value (0.50) can be partly attributed to cases where the WAS measured only a portion of the possible

isomers at a given molecular formula (e.g., isoprene and monoterpenes), although the slope (WAS vs. PTR, 0.9 ± 0.1) indicated reasonable overall agreement (Table 2). The WAS DMS EF, however, is 7-17× lower than that determined by PTR-TOFMS, despite being directly calibrated in both datasets. A recent study comparing the detection of organosulfur compounds between these two techniques demonstrated good agreement for DMS (Perraud et al., 2016); thus the reason for this discrepancy in this work is currently unknown.

The most overlapping compounds (72) were observed between GC×GC-TOFMS and PTR-TOFMS (Table 2). The compounds that were directly calibrated in both instruments are compared in Fig. 3b and include light oxygenates, aromatic compounds, and isoprene/monoterpenes. Acetone ($C_3H_6O$) and acetonitrile ($C_2H_3N$) are known to breakthrough the ATD cartridges used for GC×GC-TOFMS sample collection (Hatch et al., 2015) and thus are expectedly below the 1:1 line (outlined with gray circles, Fig. 3b). Despite the underestimation by the GC×GC-TOFMS, the EFs for acetone and

acetonitrile are linearly correlated with those determined by PTR-TOFMS (Fig. 3b). Other calibrated compounds (except monoterpenes) agree well between the two instruments, falling close to the 1:1 line (slope = 1.08 ± 0.06, $R^2$ = 0.96 not including acetone, acetonitrile, and monoterpene data points), despite application of single isomers for PTR-TOFMS calibration (Section 2.1.3).

In contrast to the other standard compounds, the monoterpene (MT, $C_{10}H_{16}$) EFs exhibited greater variability between the

two instruments. In addition to the parent ion occurring at $m/z$ 137 ($C_{10}H_{17}^+$), MTs are known to fragment following protonation in PTRMS instruments, yielding a major fragment ion at $m/z$ 81 ($C_6H_9^+$); the degree of fragmentation is isomer dependent (Maleknia et al., 2007; Tani et al., 2003; Warneke et al., 2003). The MT emission factors reported by Stockwell et al. (2015) were calibrated using $m/z$ 81 due to the high degree of fragmentation of the α-pinene standard under the PTR-TOFMS drift tube conditions utilized during FLAME-4. A comparison of the calculated MT EFs determined using $m/z$ 137

($EF_{137}$) and $m/z$ 81 ($EF_{81}$) is given in Fig. S3 and shows that $EF_{137}$ varies between ~15% to 95% of $EF_{81}$. The widest differences between $EF_{137}$ and $EF_{81}$ occurred in the fires of fuels that are not known to be MT emitters (i.e., rice straw (Kesselmeier and Staudt, 1999) and peat, Fig. S3). The high $EF_{81}$ values for such smoke samples can be partly attributed to the presence of $C_6H_8$ compounds in BB smoke, which will be detected at $m/z$ 81 upon protonation. Based on the GC×GC-TOFMS data, $EF(C_6H_8)$ is 1.5× and 16× that of $EF(C_{10}H_{16})$ in straw and peat smoke, respectively, indicating that $C_6H_8$

compounds can significantly interfere with the determination of MT EFs based on PTRMS data calibrated using $m/z$ 81. Based on this assessment, we find that PTR-TOFMS EFs calculated using $m/z$ 137 displayed better agreement with the GC×GC-TOFMS-calculated MT EFs (summed over all isomers). The mean difference between the PTR-TOFMS and GC×GC-TOFMS MT EFs improved from 1.2 g/kg to 0.93 g/kg when $m/z$ 137 was used for calibration instead of $m/z$ 81;



omitting the spruce smoke data points due to other potential interference (discussed below), the mean difference among the remaining three MT samples improved from 0.51 g/kg to 0.15 g/kg.

Despite the improved agreement using $EF_{137}$, the PTR-TOFMS MT EF remains 2.8× (spruce) and 35× (peat) higher than that measured by GC×GC-TOFMS, compared with 1.2× and 1.4× for pine and straw, respectively (Fig. 3b). Interference from

other species at $m/z$ 137 is possible and would likely vary from fuel to fuel. For example, the presence of bornyl acetate ($C_{12}H_{20}O_2$) may explain the nearly 3-fold higher MT EF in spruce smoke. Bornyl acetate has been found to compose nearly 50% of the essential oil in black spruce needles (more than all MTs combined) (von Rudloff, 1975) and is further detected at the MT masses in PTRMS measurements ($m/z$ 137 and 81) due to fragmentation and loss of $C_2H_4O_2$ (Kim et al., 2010). In addition to a small bornyl acetate EF calculated from the GC×GC-TOFMS cartridge measurements of the spruce fire (Table

S1), the qualitative GC×GC-TOFMS analysis of species desorbed from filter samples (see (Hatch et al., 2015) for details) showed that the bornyl acetate peak area was ~6× higher than the second most abundant compound observed in the spruce smoke filter samples (data not shown), indicating that significant concentrations of bornyl acetate were indeed present in spruce smoke. Thus bornyl acetate may have contributed significantly to the PTR-TOFMS MT signal in the spruce burn and the discrepancy with the GC×GC-TOFMS MT measurement; however the extent of such interference is currently unknown.

The large MT discrepancy in peat smoke is more puzzling, particularly because it is not known how much MT emissions are expected from burning peat that is derived mostly from plant matter that has decayed over hundreds of years. The peat burned here was a core sample taken from a disturbed site and likely included some non-peat fuels that may influence the potential MT emissions. A duplicate cartridge sample of the peat burn analyzed on a second column set (see Supplementary Information; data not shown) confirmed that negligible MT emissions were observed by GC×GC-TOFMS during this burn.

However, an EF of 0.43 g/kg for α-pinene + β-pinene was calculated in the peat burn from the WAS measurements, which is nearly twice as high as the PTR-TOFMS MT EF of 0.24 g/kg. We note that the WAS EF for α-pinene + β-pinene was zero for spruce smoke, where abundant MT emissions would be expected from the burning of fresh (<1 week old) boughs. A GC×GC-TOFMS measurement from the peat smog chamber experiment showed negligible MT levels, so smog chamber contamination does not appear to have played a role in the WAS measurement. Although unknown problems in the cartridge

sampling and/or analysis cannot be completely ruled out at this time, it is unlikely that MTs present in peat smoke would have gone undetected in three different cartridge samples (two room burn replicates + one smog chamber sample) during GC×GC-TOFMS analysis. Given the wide variability among these instruments for the determination of MTs and the extent to which these or similar techniques are used to measure ambient MTs, more work is clearly needed to understand the emissions of these compounds.

Regarding potential MT interference during PTRMS analysis, we additionally highlight that oxygenated compounds with nominal MW of 136 g/mol were observed by PTR-TOFMS during FLAME-4 (Stockwell et al., 2015). For peat and straw smoke, the combined EF of such compounds were ~30% and ~44% that of the MT EF, respectively, compared to ~11% (pine) and ~1% (spruce) for the conifers. Thus MT EFs determined using PTRMS instruments equipped with nominal mass resolution mass analyzers (e.g., quadrupole) could be considerably overestimated for burns of fuels that are not MT emitters.



Therefore, caution is warranted for the determination of MT EFs in smoke using PTRMS instruments due the high complexity of BB emissions.

The correlation of all overlapping data between GC×GC-TOFMS and PTR-TOFMS is given in Fig. 3a, where essentially all GC×GC-TOFMS data points are associated with PTR-TOFMS measurements due to the very limited overlap with OP-FTIR.

The agreement (slope = 0.48 ± 0.02, $R^2$ = 0.83, Table 2) among all overlapping compounds is not as robust as for the calibrated compounds. To more clearly show the range of the comparison, a histogram of the ratio of GC×GC-TOFMS EFs to PTR-TOFMS EFs is included in Fig. 3c for individual burns, as well as cumulatively for all four burns. The distribution is nearly log-normal, with a longer tail at low ratios. The geometric mean among all burns is 0.65 (geometric standard deviation = 0.42, median = 0.71); a similar distribution is observed for all fuels. The mean/median lie closer to 1 than the

slope of the correlation plot because the distribution statistics are less influenced by outliers, particularly those at high EFs. In particular, the slope of the correlation plot is significantly influenced by the high spruce MT EF determined by PTR-TOFMS (described above; Fig. 3b); when that data point was removed from the linear regression as a sensitivity test, the slope improved to 0.75 ± 0.03, in closer agreement with the histogram mean and median. This demonstrates that the GC×GC-TOFMS and PTR-TOFMS generally agreed within ~30% on average, which is within the reported uncertainties for

each measurement.

The poorer agreement between the GC×GC-TOFMS and PTR-TOFMS compared with the other instrument pairs (Table 2), can be due to multiple factors, including that quantification of uncalibrated compounds is subject to significant error. Such compounds were calibrated using surrogate standards (GC×GC-TOFMS (Hatch et al., 2015)) or mass-dependent calibration (PTR-TOFMS (Stockwell et al., 2015)). Therefore, the overall agreement could be improved by more thoroughly calibrating

the PTR-TOFMS data and directly calibrating the overlapping species detected by GC×GC-TOFMS, as indicated by the close agreement among the standard compounds (Fig. 3b). Further, polar compounds are more likely to be underestimated by GC×GC-TOFMS where significant isomers may not elute from the GC columns or may be lost to the filter used during sampling. This underestimation can be seen in Fig. 3a, where markers for GC×GC-TOFMS data points are scaled by O:C ratio (from 0-0.75 for GC×GC-TOFMS data; none of the WAS NMOGs that overlap with other instruments are oxygenated

and thus for visual clarity these markers were not scaled). Many of the compounds with relatively high O:C ratio fall below the 1:1 line, highlighting the general underestimation of oxygenated compounds by GC×GC-TOFMS. Thus, more work is needed to understand and optimize the GC×GC-TOFMS sampling and analysis methods to characterize polar compounds in BB emissions.

### 3.3 Emissions Characterization

Discrepancies among the instruments were generally well understood and provided sufficient confidence in the data to construct emission profiles. Figures 4 and 5 show the overall gas-phase composition including all measurements for peat and straw smoke, respectively, sorted into major chemical classes; analogous figures for pine and spruce smoke are shown in Figs. S4 and S5. The synthesized EF database is included in Table S1. Although furans are aromatic compounds, they are



treated as a separate class; 'aromatic' in this paper therefore refers to benzene derivatives. Unknown compounds in the PTR-TOFMS dataset were categorized based on the number of double-bond equivalents (i.e., compounds with DBE ≥4 were assigned as aromatic); such compounds, particularly oxygenates, are included in the 'unknown/double counting category', due to the lack of information regarding functional groups. This category includes compounds for which both PTR-TOFMS

and GC×GC-TOFMS or WAS data were kept (Section 2.3). These cases either reflect an incorrect calibration or sampling artifact in one or both instruments (leading to double counting) or unknown compounds unaccounted for by the GC techniques. Therefore, the unknown/double counting segments are most likely to be revised or re-classified by future measurements. The total number of compounds observed per fuel following data reduction ranged from 467 (peat) to 569 (pine), including isomers and a few potentially double counted compounds (Table S1). For comparison, the number of

unique chemical formulas ranged from 164 (peat) to 180 (straw), demonstrating both the diversity of compounds emitted from BB, but also the large number of isomers detected by the GC techniques (Fig. 2b).

NMOG profiles for straw (Fig. 5), pine (Fig. S3), and spruce (Fig. S4) are similar, with the largest contribution from oxygenated aliphatic compounds followed by aliphatic HCs. Recently (Gilman et al., 2015) determined that oxygenated NMOGs constituted 57-68% of all BB emissions compiled from GC-MS, OP-FTIR and a variety of chemical ionization

mass spectrometer measurements from laboratory burns of fuels common to different regions of the United States. The percentage of all oxygenated NMOGs for pine and straw smoke determined in this work was similar at 55% and 54% of the NMOG EF, respectively. The oxygenates in spruce smoke composed only 43% of the total NMOG EF, partly due to the very high MT emissions measured by PTR-TOFMS (Fig. S5 and Section 3.1.2). Further, oxygenates constituted only 25% of the emissions in peat smoke, which was dominated by aliphatic HCs (57%, Fig. 4). In all smoke samples, compounds

with CN ≤ 3 constitute 40-50% of the total NMOG EF, largely due to ethene, methanol, formaldehyde, acetaldehyde, and acetic acid (Figs. 4-5, S4-S5).

### 3.3.1 Volatility

The $C^*$ of all measured NMOGs was estimated using the parameterization of (Li et al., 2016) described in section 3.1. The compounds are displayed in molecular corridors in Fig. 1c and highlight that a large number of HC and oxygenated IVOCs

were detected (IVOCs defined as $C^* = 10^3 - 10^6$ µg/m³ (Donahue et al., 2009)). Approximately 65 unique molecular formulas (range 61-68) were measured in the IVOC range. Except for organonitrates, which are likely misclassified as IVOCs using this approach, all IVOCs determined in FLAME-4 were measured solely by PTR-TOFMS and GC×GC-TOFMS. In all cases, the PTR-TOFMS measured a higher fraction of IVOCs than GC×GC-TOFMS (Table 1), likely due in part to the use of a heated sample inlet with the PTR-TOFMS measurements, which provides improved transmission of

lower volatility compounds compared to the room temperature sample line and filter used for cartridge sampling (Sections 2.1.3 and 2.1.4). Based on the applied $C^*$ parameterization and volatility classifications, no SVOCs were detected with the analytical methods applied in this work (Fig. 1c). It is expected that with the high OA concentrations (~1000-6000 µg/m³) during these burns much of the SVOC was present in the condensed phase; additionally SVOCs may have been lost to





surfaces present in the combustion chamber (e.g., as has been modeled by Bian et al. (2015) for smog chambers). As seen in Fig. 1a, there are a few publications for which SVOCs in gaseous BB emissions (e.g., MW > ~250 for HCs) have been reported (Garcia-Hurtado et al., 2014; Hays et al., 2002; Schauer et al., 2001). However, more work is needed to better identify and quantify the semi-volatile components of BB smoke.

To further probe the fraction of the NMOG EF attributable to IVOCs, all NMOGs were binned by estimated $C^*$. The resulting EF distribution as a function of volatility is included in Fig. 6 for pine smoke; analogous figures for the other fuels are included in the Supplementary Information (Figs. S6-S8). The volatility of compounds measured across all four instruments during FLAME-4 spans 9 orders of magnitude; 7 of these bins contain significant mass. In the pine smoke sample, IVOCs accounted for ~11% of the total NMOG EF (6-8% for the other fuels; Table 1); the majority of which falls at

the high end of the IVOC volatility range (i.e., log$C^*$ ~5-6; Fig. 6, Figs. S6-S8). For comparison, the compounds typically measured in BB smoke (based on Table 1 of (Akagi et al., 2011)) and those included in the EPA SPECIATE emission inventory (EPA, 2008) are also included. Because the EFs (or compound weighting) of these two compilations are based on an ecosystem average (e.g., temperate forest) whereas the FLAME-4 data are based on a single burn of a single fuel, comparison of EF values among these studies is not very meaningful. Rather, we emphasize the portion of FLAME-4

emissions that would have been observed if only the routine compounds had been measured; thus for each compound included in (Akagi et al., 2011) or the SPECIATE inventory we have applied the corresponding EF from the combined FLAME-4 dataset (Table S1).

The volatility of the compounds in both (Akagi et al., 2011) and SPECIATE spans 8 orders of magnitude; however compounds in only 5 bins contribute significantly to the overall EF in both cases (Fig. 6). The compounds included in the

SPECIATE database and (Akagi et al., 2011) account for 63% and 66%, respectively, of the total NMOG EF detected here, leaving more than 30% of the NMOG EF unaccounted for in pine smoke (Fig. 6). Akagi et al. (2011) was based primarily on field measurements deemed representative for major BB types. They estimated that about 50% of the NMOG mass was unknown based on PTRMS spectra of lab-generated smoke available at the time and provided estimates of unmeasured/unidentified NMOG, however they were not speciated. This work now identifies and quantifies a large fraction

of the unknown mass highlighted in that compilation. The fraction of each bin accounted for by the routinely measured compounds or SPECIATE inventory decreases with decreasing volatility (Fig. 6). Thus if the weighting values from SPECIATE are used, the total EF would be mapped to a group of compounds with a significantly higher mean volatility. In particular, IVOCs were almost entirely absent (Fig. 6) based on the applied volatility parameterization; less than ~1% of the IVOC EF measured in this work for pine smoke was accounted for by the compounds included in the Akagi et al. (2011)

compilation (based primarily on field studies) and the SPECIATE inventory. This is likely a conservative estimate for the fraction of unspeciated emissions given that the largest underestimation occurs at the lower volatility end of the distribution (Fig. 6), where some fraction of the compounds were also missed by the analytical techniques used in this work. In particular, because smoke collects in the combustion chamber during room burn experiments, losses of sticky or lower volatility compounds to surfaces or particles can occur (Stockwell et al., 2014).





The distribution of measured IVOCs among the major chemical classes is shown in Fig. 7. For all burns except peat, oxygenates are overwhelmingly dominant, accounting for over 75% of the IVOC emissions. However, the influence of different oxygenated classes varied from fuel to fuel, with oxygenated aromatics constituting nearly 70% of the IVOC EF in pine smoke. These compounds were primarily measured by PTR-TOFMS, which thus explains the very large difference in

the fraction of IVOCs measured by PTR-TOFMS and GC×GC-TOFMS for pine smoke (Table 1). IVOCs from straw and spruce include a higher relative fraction of furans and oxygenated aliphatics (which was mostly bornyl acetate in spruce smoke). In contrast, only 53% of the IVOCs detected in peat smoke were oxygenated. IVOCs in this burn comprised a higher fraction of aromatic and aliphatic HCs than observed in other fuels (Fig. 7). The high fraction of oxygenated IVOCs in BB emissions stands in stark contrast to IVOCs emitted from fossil-fuel combustion, which has generally been measured

as (or assumed to be) almost entirely hydrocarbons, particularly alkanes (Zhao et al., 2014; Presto et al., 2009; Tkacik et al., 2012). Our FLAME-4 measurements, however, did not include gas-phase measurements of polycyclic aromatic hydrocarbons (PAHs) larger than acenaphthylene, which have been widely measured in BB emissions (Dhammapala et al., 2007; Hall et al., 2012; Hays et al., 2002; Jenkins et al., 1996; Schauer et al., 2001; Singh et al., 2013). In pine smoke, for example, Schauer et al. (2001) reported a total EF for gaseous PAHs larger than acenaphthylene of 0.045 g/kg, which is ~1%

of the total IVOC EF measured from pine smoke in this work.

### 3.3.2 SOA Yields

To model BB SOA formation, the propensity of observed compounds to form SOA needs to be known. In most widely used models, SOA formation is based on SOA yields (mass of SOA formed/mass of precursor reacted) determined from smog chamber studies (e.g., as described in (Barsanti et al., 2013). An alternative approach is to use a semi-explicit gas-phase

chemical mechanism to predict the oxidation products of individual NMOG precursors and calculate the gas/particle partitioning of the oxidation products directly. This latter approach was applied by Derwent et al. (2010), who determined the SOA formation potential of 113 anthropogenic NMOGs using the Master Chemical Mechanism v3.1; SOA formation potentials were reported relative to toluene. Recently Gilman et al. (2015) used the model-derived SOA potentials from Derwent et al. (2010) to evaluate potential SOA formation from BB emissions. Aromatic compounds contributed most of

the SOA formation potential from BB emissions: 18-41% from aromatic hydrocarbons and 50-75% from oxygenated aromatic compounds (e.g., benzaldehyde and phenol derivatives) depending on the fuel. The SOA formation potential from monoterpenes using this approach was notably low (factor of 5 less than toluene); (Gilman et al., 2015) conducted a sensitivity study and determined the monoterpene contribution was still minimal even when the SOA yield potential for monoterpenes was increased 10-fold.

In the study by Gilman et al. (2015), <37% of the compounds overlapped with those reported in (Derwent et al., 2010); thus assumptions had to be made regarding representative compounds (and thus representative SOA formation potentials) for nearly two-thirds of the compounds relevant for BB. The majority of the non-aromatic compounds were assigned SOA formation potentials ≤1% that of toluene. Ideally modeled SOA formation potentials would be available for the specific



compounds of interest and those SOA formation potentials would be compared with smog chamber SOA yield data. For the compounds measured in this work, an extensive literature search was performed to determine the extent of published SOA yield data. For the top 100 compounds from each fuel, which account for ~90% of the total NMOG EF for each fuel (87-91%), the measured EF was scaled by the corresponding rate constant for reaction with OH to emphasize the most reactive

compounds and by carbon number as a rough proxy for potential SOA contribution. These scaled EFs are hereafter termed 'reactive carbon'. Measured OH rate coefficients were used where available (Calvert et al., 2015), otherwise values were estimated using the EPA's estimation program AOPWIN (v1.92, U.S. EPA Estimation Programs Interface Suite, 2014), a tool that is based on standard structure-reactivity relationships (Atkinson, 1987; Kwok and Atkinson, 1995). Although a few unknown compounds were present in the top 100 compounds, they were not included in this analysis due to the inability to

estimate reasonable OH reaction rate constants. Assuming a generic rate constant of $1\times10^{-11}$ cm$^3$ molecules$^{-1}$ s$^{-1}$, the unknown compounds contributed less than 5% to the total reactive carbon of the top 100 compounds and thus their omission should not significantly impact the results. We have also omitted the PTR-TOFMS-derived MT EF for spruce smoke due to the suspected interference of bornyl acetate (see section 3.2.2). Compounds were then sorted by the number of publications reporting an SOA yield via OH-radical oxidation (as of May 2016); classifications and corresponding literature references

are provided in Table S2. Results are shown in the Fig. 8 and 9 pie charts for pine and straw smoke, respectively (Figs. S9 and S10 for spruce and peat smoke), illustrating that only 12-22% of the reactive carbon is associated with very well studied compounds (5+ publications). Such compounds include toluene, *m*-xylene, α-pinene, and isoprene. In contrast, between 55% (pine) and 77% (straw) of the reactive carbon is associated with compounds for which SOA yields are unknown or understudied (0-1 publications). These fractions could increase appreciably if the neglected unknown compounds are

significantly more reactive than assumed above, as SOA yields likely have not been assessed for compounds that could not be identified in this work. Of the understudied compounds, those most likely to form SOA following reaction with OH radical are outlined in gray in the pie charts of Figs. 8, 9, S8, and S9. These understudied potential precursors constitute between 22% (peat) and 56% (straw) of the included reactive carbon for each burn. Therefore, even with improved speciation measurements, critical data for modeling BB SOA formation are missing for a significant fraction of the

potentially reactive material.

Many of the understudied potential precursors are furan derivatives and polyunsaturated aliphatic hydrocarbons; only ~10% (peat) to 28% (straw) of the reactive carbon contributed by understudied precursors is attributed to aromatic compounds. Thus the largest gaps in known SOA yields relevant for BB are associated with non-aromatic compounds (furans notwithstanding). To better identify specific candidates for future smog chamber studies, the top 10 understudied potential

precursors are shown in the corresponding bar charts as a percentage of the reactive carbon included in the gray-outlined wedge (Figs. 8, 9, S9, and S10). For all four fuels, furan derivatives account for 3-5 of the top 10 understudied compounds. Furfural, 2-methyl furan, 2-furan methanol, and 2-hydroxy-3-methyl-2-cyclopentenone (tentatively identified by PTR-TOFMS) are common to the top 10 lists for all fuels, although 1,3-cyclopentadiene is present for three of the burns. Given





the ubiquity and potential importance of these compounds, future smog chamber experiments with these species may significantly help to narrow knowledge gaps regarding SOA yields of BB emissions.

**4 Conclusions**

Data collected from a unique combination of four instrumental approaches deployed during FLAME-4 have been compared
to evaluate the compositional space and calculated EFs accessed by each instrument and to provide comprehensive BB gas-phase emissions profiles for four sampled fuels. OP-FTIR has the least amount of sampling artifacts, but very limited ability to probe high MW species. PTR-TOFMS with a heated sample line may be best for detecting the lowest volatility and most polar compounds, but has significant limitations for compound quantification and identification, and additionally is unable to detect saturated hydrocarbons. We further found evidence for significant interference in the determination of monoterpene
EFs by PTR-TOFMS due to bornyl acetate in spruce smoke and by $C_6H_8$ compounds from non-monoterpene emitting fuels. As a result, monoterpene EFs calculated using the protonated parent ion (*m/z* 137) displayed better, though still variable, agreement with the cumulative GC×GC-TOFMS-derived monoterpene EF than the commonly used fragment ion (*m/z* 81). GC×GC-TOFMS can speciate numerous various isomers, but sticky compounds or compounds that breakthrough cartridges may not be detected or may be underestimated. WAS does not suffer from breakthrough, but is limited to relatively more
volatile compounds than cartridge sampling. The major findings of the data analysis are: 1) all of these techniques together were able to positively or tentatively identify the compound structures for 87-92% of the NMOG EF detected in smoke sampled during FLAME-4 with the remaining EF assigned chemical formulas; 2) a general comparison shows that despite some outliers for specific species or fires, the overall agreement for overlapping species is within the uncertainty (< ~30%) for any given technique with no large bias evident; 3) this allows us to further conclude that each technique contributes a
distinctive ability to identify some important subset of the total BB-derived NMOG. Application of a range of instruments is therefore currently necessary for adequately measuring the wide variety of compounds emitted from BB.

Deployment of this suite of instruments during FLAME-4 enabled us to construct a comprehensive database of emission factors for compounds that cover a wider volatility range than traditionally measured. Although light compounds (carbon number ≤ 3) constituted 40-50% of the total NMOG EF, a significant fraction (6-11%) of the observed BB emissions were
attributed to IVOCs, which are generally unaccounted for using the typical measurement approaches. These lower volatility compounds may be efficient SOA precursors. Further, assessment of BB-relevant SOA yields showed that <25% of NMOG emissions can be attributed to compounds with well-characterized SOA yields. Instead, 20-56% of the reactive carbon was attributed to understudied compounds with the potential to form SOA, of which furan derivatives and polyunsaturated hydrocarbons dominated. Future work is therefore needed to assess the SOA-formation potential of some major compounds
emitted during BB. Ideal candidates for future smog chamber experiments were identified as a starting point for improving the scientific understanding and estimations of SOA production in smoke plumes.





### Acknowledgements

This work was supported by the National Science Foundation (Grants AGS-PRF-1231128 and ATM-0936321), NASA Earth Science Division (Award NNX14AP45G), and BLM Joint Fire Science Program (Grant L14AC00160/L16AC00005). Dr. Betsy Stone and Thilina Jayarathne (University of Iowa) are thanked for providing the organic aerosol concentrations.

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



**Table 1: Total EF (in g/kg fuel burned) and the percentage of the total EF measured by each instrument for different classes of compounds.**

| Fuel | Instrument | All Compounds | NMOG | IVOC |
|---|---|---|---|---|
| Ponderosa Pine | Total EF (g/kg) | 1780 | 36.5 | 3.96 |
| | OP-FTIR (%) | 99 | 36 | 0.0 |
| | WAS (%) | 5.2 (98) | 26 | 0.2 |
| | GC×GC/TOFMS (%) | 0.7 | 31 | 17 |
| | PTR-TOFMS (%) | 1.1 (1.6) | 55 (79) | 88 |
| Black Spruce | Total EF | 1820 | 37.3 | 2.31 |
| | OP-FTIR | 99 | 28 | 0.0 |
| | WAS | 99 | 18.0 | 0.3 |
| | GC×GC/TOFMS | 0.48 | 23 | 27 |
| | PTR-TOFMS | 1.2 (1.6) | 59 (76) | 79 |
| Indonesian Peat | Total EF | 2030 | 53.1 | 4.00 |
| | OP-FTIR | 98 | 21 | 0.0 |
| | WAS | 13 (99) | 58 | 0.9 |
| | GC×GC/TOFMS | 0.7 | 28 | 45 |
| | PTR-TOFMS | 0.7 (1.3) | 29 (50) | 59 |
| Chinese Rice Straw | Total EF | 1500 | 9.53 | 0.67 |
| | OP-FTIR | 99 | 32 | 0.0 |
| | WAS | 4.0 (99) | 30 | 0.1 |
| | GC×GC/TOFMS | 0.2 | 35 | 37 |
| | PTR-TOFMS | 0.3 (0.4) | 47 (68) | 84 |



**Table 2: Linear regression statistics for each instrument pair. For all linear regressions, the *y*-intercept was forced through zero.**

| Instrument Pair | # Overlapping Molecular Formulas | Slope | $R^2$ |
|---|---|---|---|
| WAS, OP-FTIR | 6 | 1.01 ± 0.001 | 1.0 |
| WAS, PTR-TOFMS | 12 | 0.9 ± 0.1 | 0.50 |
| WAS, GC×GC-TOFMS | 14 | 1.32 ± 0.08 | 0.82 |
| GC×GC-TOFMS, PTR-TOFMS | 72 | 0.48 ± 0.02 | 0.83 |
| GC×GC-TOFMS, OP-FTIR | 1 | - | - |



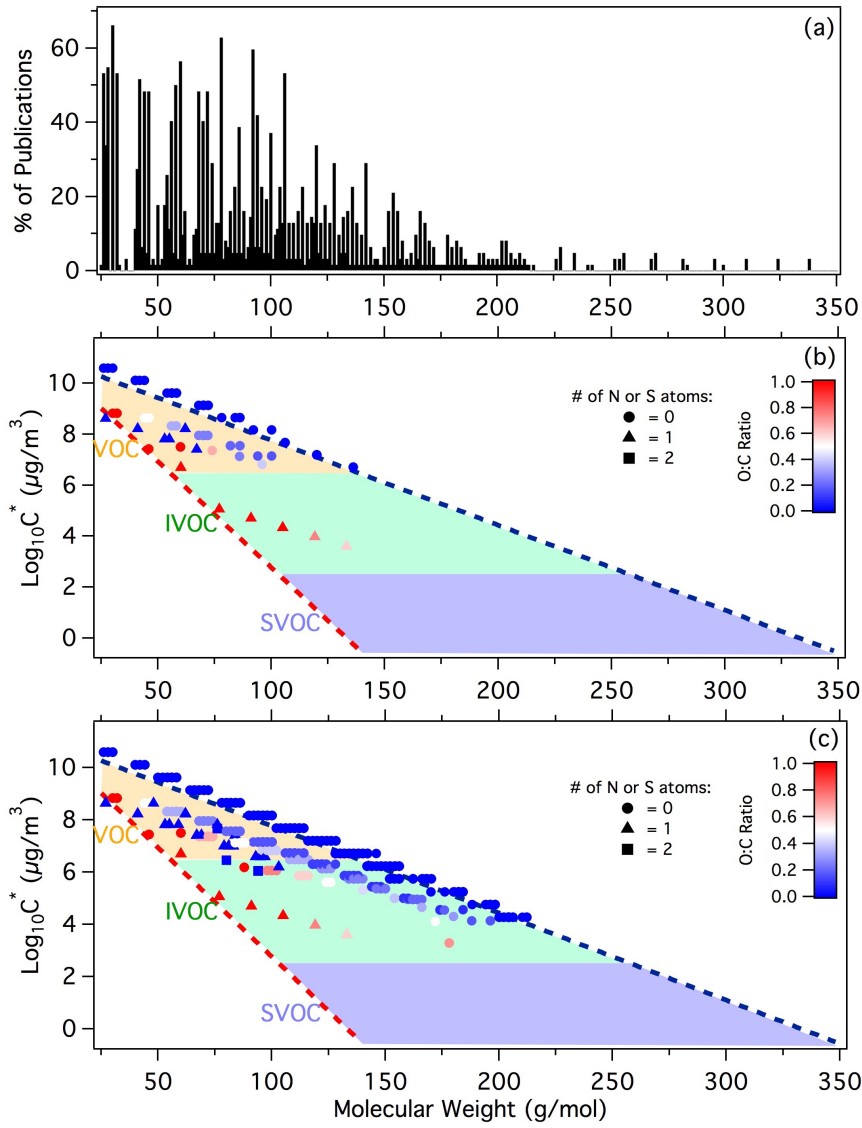

**Figure 1: (a) Percent of all relevant publications reporting biomass burning emissions of species at a given molecular weight; (b) Molecular corridors representing volatility vs. molecular weight of typically measured NMOGs (Akagi et al., 2011) based on the volatility parameterization of (Li et al., 2016). The approximate ranges for volatile, intermediate volatility, and semi-volatile compounds (as defined by (Donahue et al., 2009)) are indicated by the shaded regions; (c) As in (b), for the compounds measured in this work from all fuels. In panels (b) and (c), the colorscale saturates at an O:C ratio of 1.**




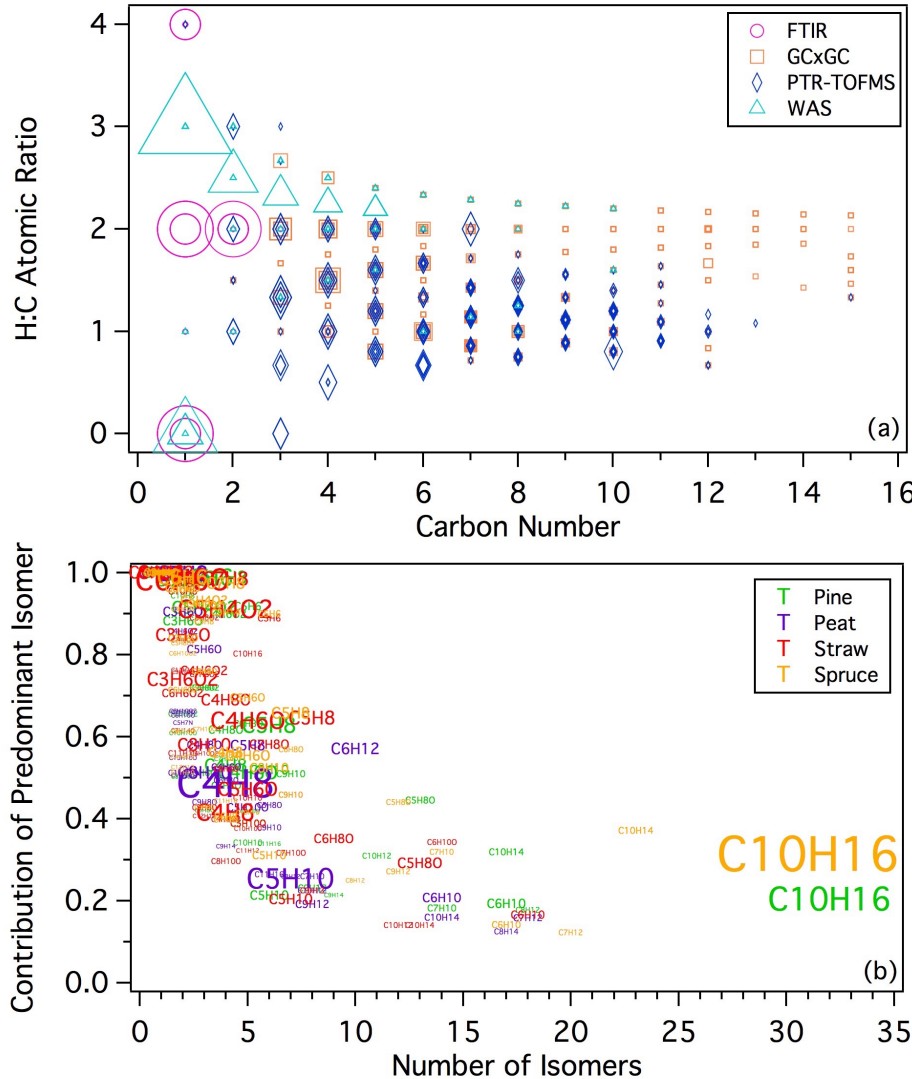

**Figure 2: (a)** Comparison of the range of compounds measured by each instrument as a function of H:C ratio and carbon number. Marker size is proportional to the O:C ratio. Data from all four burns are represented; **(b)** Contribution of the predominant
5    isomer as a function of the number of observed isomers. Marker size is proportional to the contribution of each isomer group to the total NMOG EF (from 0-5%).



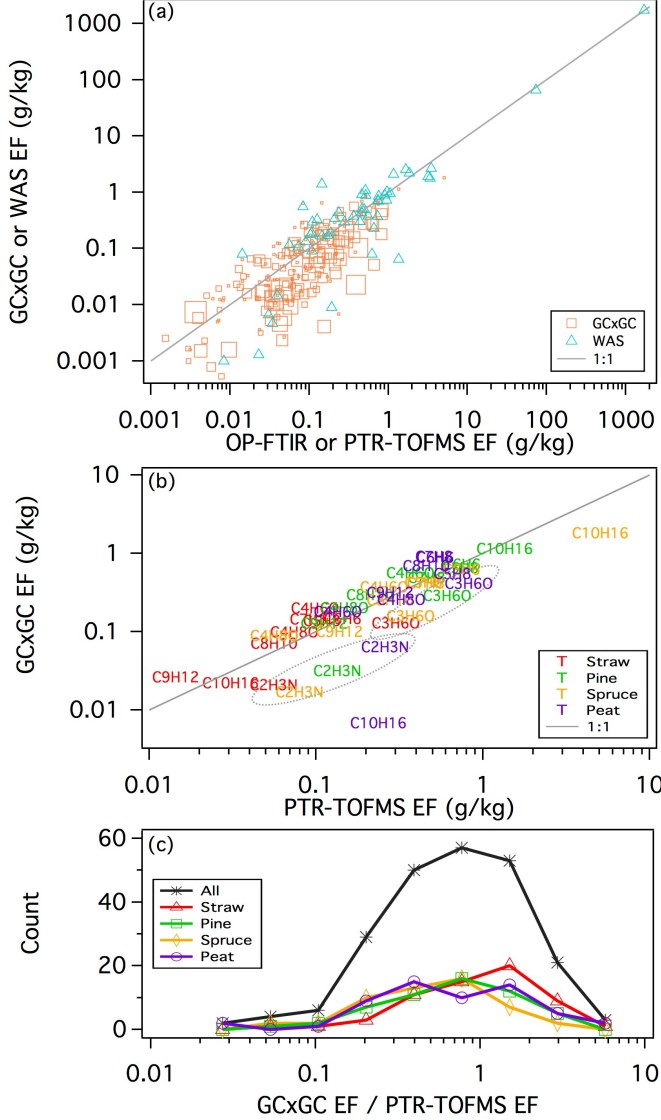

**Figure 3.** **(a) Comparison of the emission factors determined by GC×GC-TOFMS or WAS (*y*-axis) with those measured by PTR-TOFMS or OP-FTIR. Marker size is proportional to O:C ratio for GC×GC-TOFMS comparisons only; (b) Comparison of GC×GC-TOFMS and PTR-TOFMS emission factors determined for overlapping standard (i.e., calibrated) compounds only.**
5     **Dotted gray circles denote compounds affected by known breakthrough artifacts during cartridge sampling; (c) Histogram of the ratio of GC×GC-TOFMS emission factors relative to PTR-TOFMS emission factors for all overlapping compounds within individual burns and summed over all burns.**





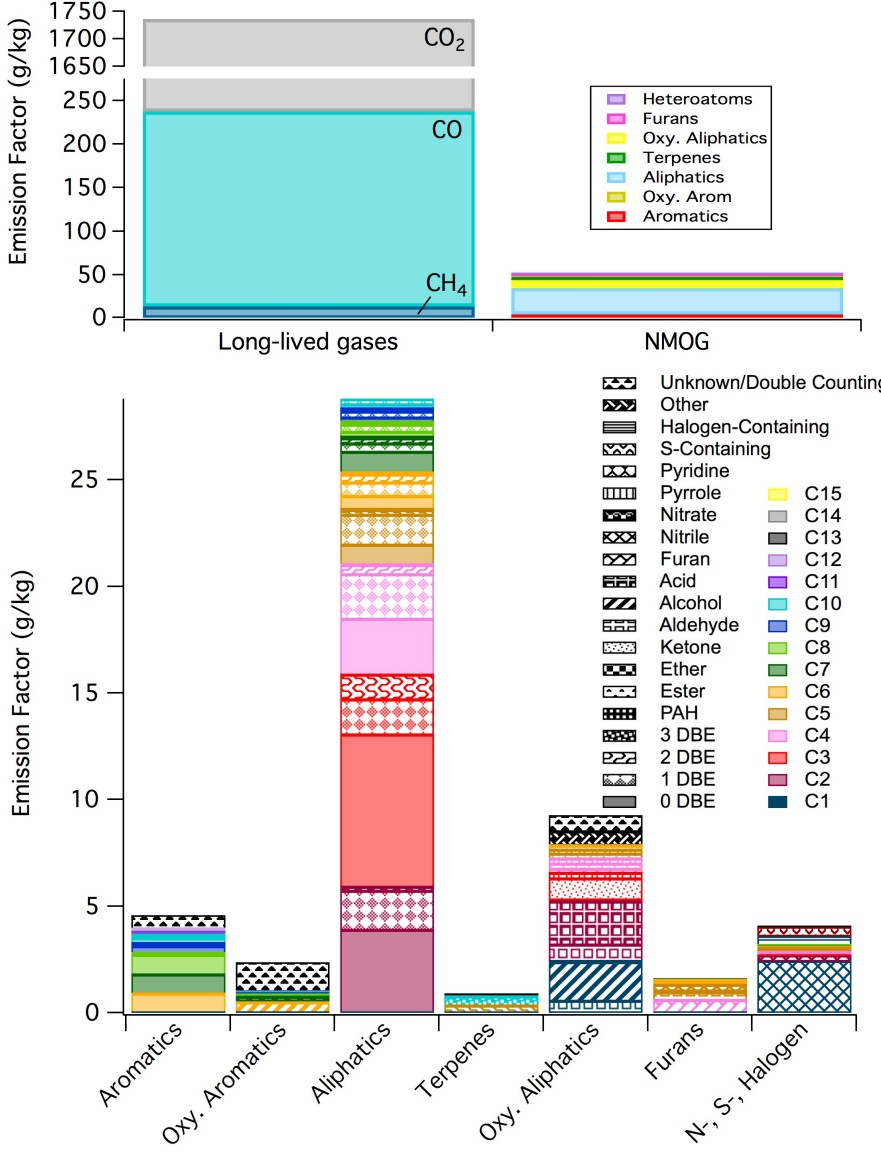

**Figure 4: Gas-phase emission factors from an Indonesian peat fire. Top panel: long-lived gases compared to NMOG. Bottom panel: Speciation of NMOG; colors represent carbon number and patterns indicate functionality. 'DBE'= double bond equivalents.**



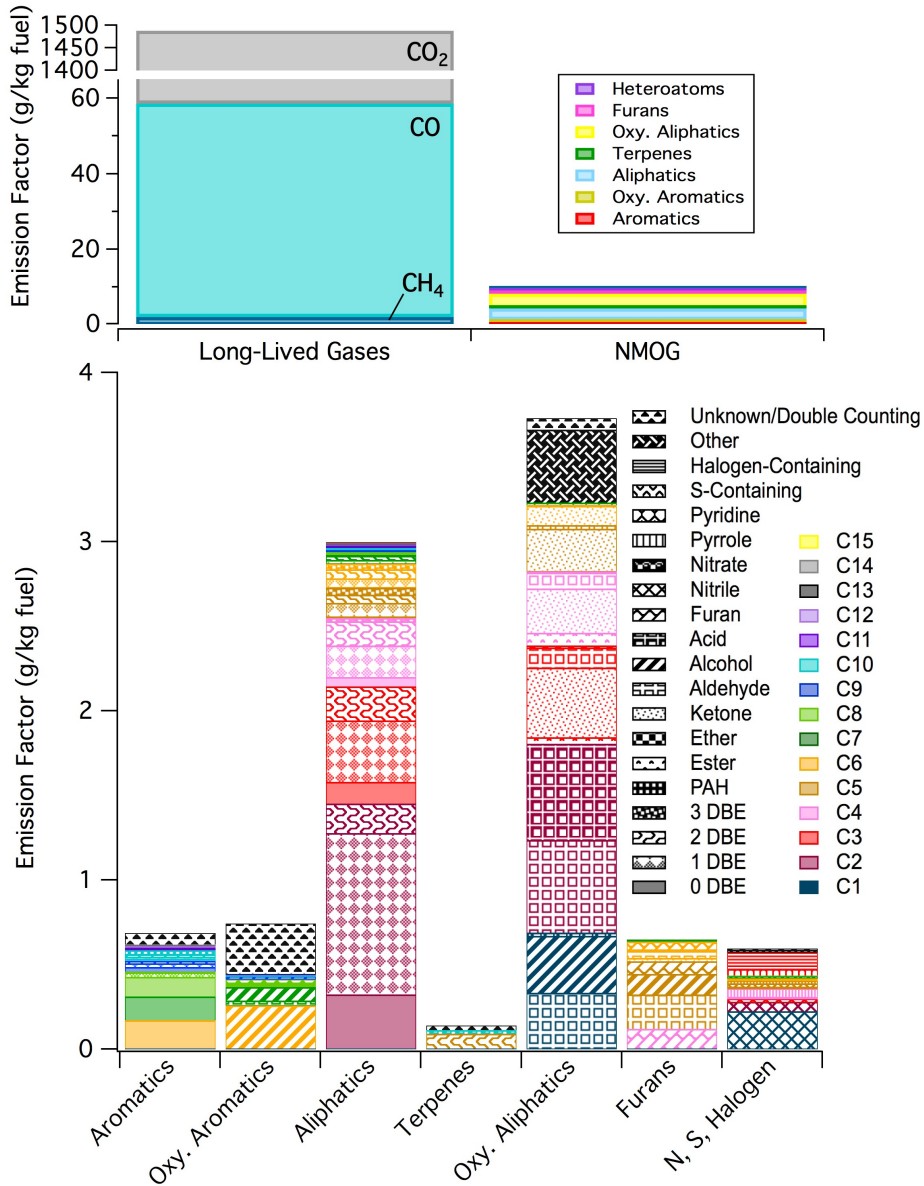

**Figure 5: As in Fig. 3, for a Chinese rice straw fire.**





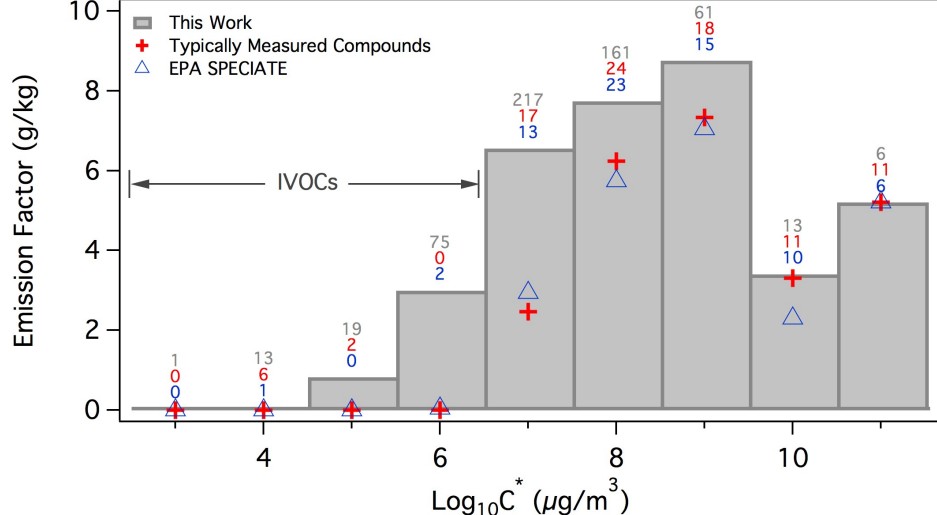

**Figure 6: Emission factors of NMOG determined in pine smoke, as a function of volatility (see text). Red (+) and blue (Δ) markers indicate the contribution from typically measured compounds based on (Akagi et al., 2011) and the EPA SPECIATE emission inventory (EPA, 2008), respectively. The number of compounds included in each bin is indicated above the bars.**





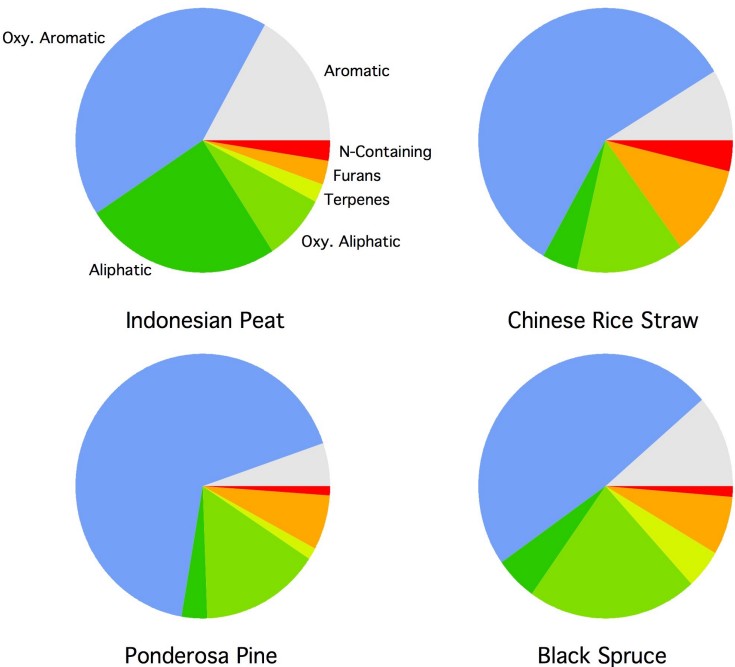

**Figure 7: Distribution of intermediate volatility compounds among the major compound classes.**



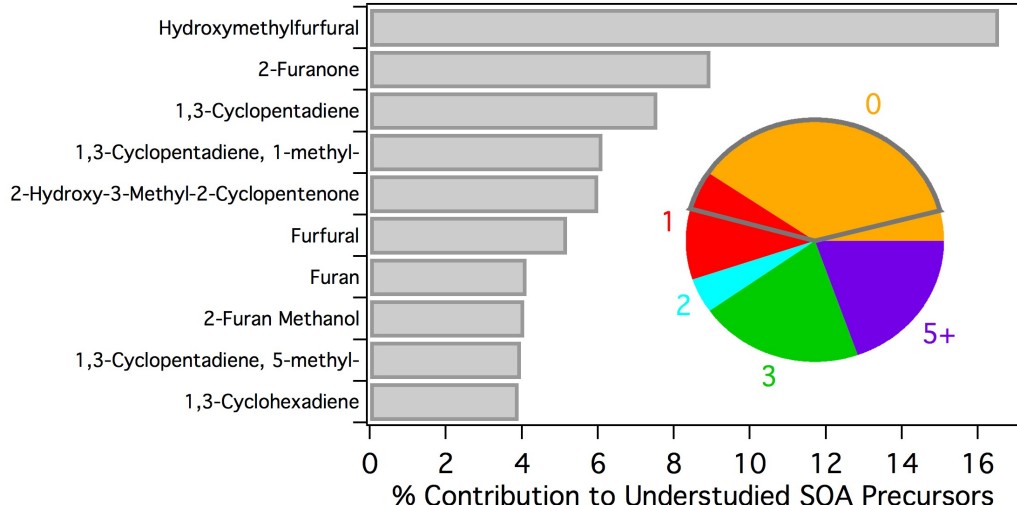

**Figure 8: Assessment of SOA yields for compounds detected in the ponderosa pine fire. Pie chart: Classification of reactive carbon (see text) by the number of publications reporting an SOA yield following hydroxyl radical oxidation. The gray-outlined wedge represents the understudied compounds with the greatest potential to form SOA. Bar chart: Percent contribution of the top 10 compounds included in the gray wedge.**

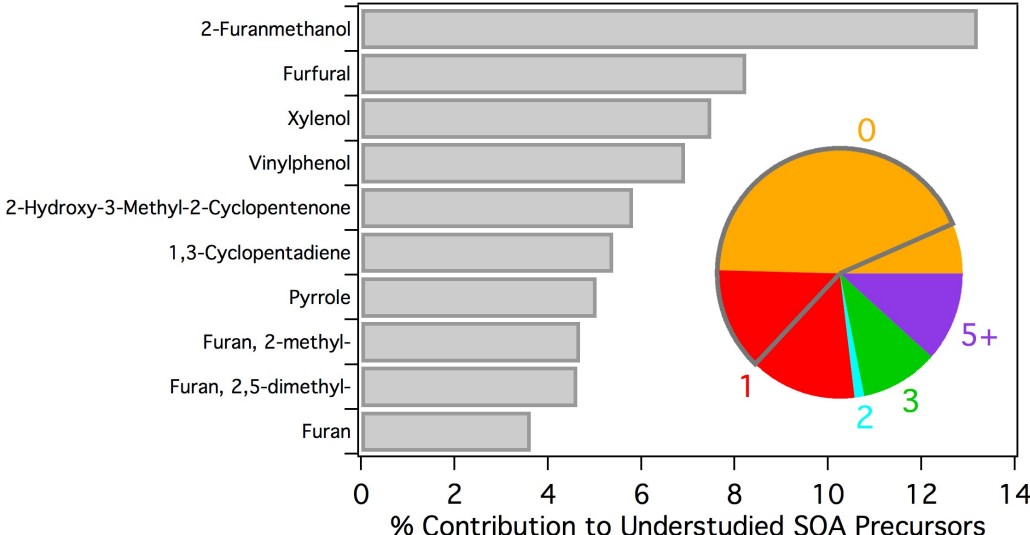

**Figure 9: As in Fig. 8, for a Chinese rice straw fire.**