# Peer review of "Multi-instrument comparison and compilation of non-methane organic gas emissions from biomass burning and implications for smoke-derived secondary organic aerosol precursors"

_Atmospheric Chemistry and Physics, 2016_

## Referee Comment (RC1) · Anonymous Referee #1 · 12 Aug 2016

This paper reports laboratory measurements of biomass burning (BB) emissions taken during the Fire Lab at Missoula Experiment (FLAME-4). Fresh BB emission were characterized using four complimentary techniques: open-path FTIR (OP-FTIR), proton-transfer-reaction time-of-flight mass spectrometry (PTR-TOFMS), two-dimensional gas chromatography time-of-flight mass spectrometry (GC×GC-TOFMS), and whole air sampling (canister) with gas chromatography-mass spectrometry (GC-MS). FLAME-4 was the first application of PTR-TOFMS and GC×GC-TOFMS for laboratory BB emissions. FLAME-4 results from the first three techniques have been reported previously (OP-FTIR, Stockwell et al., 2014; GC×GC-TOFMS, Hatch et al., 2015; PTR-TOFMS,

[Figure]

Stockwell et al., 2015). This paper synthesizes data four the four techniques to 1) compare the compositional space accessed and emission factors (EF) measured by the techniques, 2) provide comprehensive emission profiles for four fuels (Ponderosa Pine, Black Spruce, Indonesian Peat, and Chinese Rice Straw) and 3) characterize the volatility distribution of the emissions and identify potentially important secondary organic aerosol (SOA) precursors which have been understudied (e.g. in photochemical chamber experiments of SOA formation).

This paper present new and very valuable findings for biomass burning and atmospheric chemistry fields. The study quantifies emissions for 10's of compounds that a have significant potential for SOA formation, but have received little or no attention in previous studies of SOA. By doing so this paper identifies potentially important gaps in our current understanding of BB impacts on SOA. The careful 4 instrument intercomparison presented in the paper may be useful for reinterpreting previous studies of BB that did not benefit from the unprecedented composition coverage and diversity of methods used here. I recommend this paper for publication pending consideration of the following comments.

Comments

Please provide details on PP and BS fuels. Were they both boughs? How fresh? Was combustion limited to mostly needles and fine branch wood?

Pg 4 line 19. "chemical perturbation" does this include lamps? Was the chamber dark during influx of emissions and prior to acquisition of WAS?

Figure 2b is messy and difficult to read. I suggest modifying to include only a handful of species that coincide with the main points of the corresponding text (P9, L24 – P10 L5)

Any thought on why the high contribution of alkanes for peat versus other fuels?

P11, L19-33: Does the significant interference of C6H8 compounds at m/z 81 identified

in this study suggest the previously reported MT EF using PTRTOFMS (Stockwell et al., 2015) should be revised? "The MT emission factors reported by Stockwell et al. (2015) were calibrated using m/z 81 due to the high degree of fragmentation of the $\alpha$-pinene standard under the PTRTOFMS drift tube conditions utilized during FLAME-4."

P14, L33: The author not the OA concentrations were 1000-6000ug/m3. Where do these numbers come from? Are these form previously published FLAME-4 papers? Please explain the origin of the OA measurements.

---

## Referee Comment (RC2) · Anonymous Referee #2 · 25 Sep 2016

Review of "Multi-instrument comparison and compilation of non-methane organic gas emissions from biomass burning and implications for smoke-derived secondary organic aerosol precursors" by Hatch et al.

The authors present emission factors for 4 different types of biomass based on laboratory measurements using 4 different methods to provide a greater coverage of organic gases in terms of mass and type. Methods are inter-compared and composite emission factors and speciation profiles are provided for the different biomass types. This is useful work for the fire science community and should be published; however, several

suggestions are made that would make this manuscript more broadly useful toward modeling wildland fire atmospheric impacts.

Emission factors for biomass burning are known to vary by combustion component (e.g., flaming to smoldering). It would be useful for the authors to describe the combustion efficiency of the samples as a function of measured CO and CO2 or if that is not available as part of the experiment some qualitative description about the relative amount of time the biomass samples were flaming or smoldering. This will allow for a more specific application of these emission factors in modeling in the future as they may be more relevant for either the flaming or smoldering component of the various biomass types included as part of this assessment.

From a modeling perspective, particle phase measurements would ideally be collected at the same time and at the same conditions as the gas phase measurements. Biomass burning clearly emits organic gases that are known to be a source of SOA in the atmosphere. However, the authors ignore ambient based field study data in their discussion of SOA formation potential that suggests little or no SOA is formed from biomass burning: Jolleys et al., 2012, ES&T, Forrister et al., 2015,GRL, Yu et al., 2016, JGR, Liu et al., 2016, JGR, and Cubison et al., 2011, ACP. This work is intended to provide emission factors and not be a chamber study looking at SOA formation so this comment is not meant to require significant changes, just to recognize that SOA formation from biomass burning in the atmosphere based on field study data is highly variable in terms of context. Perhaps IVOC emissions from biomass burning may not be that important for understanding SOA. However, future work should include measuring both gas (via the multi-instrument approach suggested here) and particle phases simultaneously to accounting for the elusive missing mass (capturing it regardless of phase).

Specific comments:

At the OA mass loadings the measurements were made at (1000-6000 ug/m3, p. 14

line 32), 50-85% of the lowest volatility IVOC they measure ($C^*=10^3$) would be in the particle phase (or 9-37.5% of $C^*=10^4$). While they didn't measure much mass at the lowest volatility, partitioning to the particle phase would have an impact on their results.

Including the IVOC EF in Fig 7 for each fuel type would help provide context on how the magnitude and composition of IVOC emissions changes across the fuels.

In the instrument to instrument comparison, the authors note difficulty estimating monoterpene emission factors due to interference from other compounds and the need to account for fragmentation. The novel approach presented here would seem to include sesquiterpenes which are not specifically discussed even though these compounds have notable SOA yields. Did these approaches do well at capturing sesquiterpenes? Are there similar fragmentation issues related to appropriately characterizing the emissions of of sesquiterpenes? There are not a lot of C15 compounds for these 4 biomass types (only 3 seem to have C15) but perhaps the sesquiterpenes are captured here as fragments.

In the SOA yields section, the authors note that furans are an important class of understudied organic gases in terms of SOA yield. In terms of this information being used for modeling atmospheric aerosol formation, is it possible that furans are chemically converted to traditional SOA precursors at time scales much faster than time steps in most operational air quality models and could simply be treated as a traditional SOA precursor?

---

## Author Comment (AC1) · 2 Dec 2016

We thank the reviewer for the thoughtful comments. Their comments are reproduced here (R), with our responses noted (AC).

Anonymous Referee #1

This paper reports laboratory measurements of biomass burning (BB) emissions taken during the Fire Lab at Missoula Experiment (FLAME-4). Fresh BB emission were characterized using four complimentary techniques: open-path FTIR (OP-FTIR), protontransfer-reaction time-of-flight mass spectrometry (PTR-TOFMS), twodimensional gas chromatography time-of-flight mass spectrometry (GC×GC-TOFMS), and whole air sampling (canister) with gas chromatography-mass spectrometry (GC-MS). FLAME-4 was the first application of PTR-TOFMS and GC×GC-TOFMS for laboratory BB emissions. FLAME-4 results from the first three techniques have been reported previously (OP-FTIR, Stockwell et al., 2014; GC×GC-TOFMS, Hatch et al., 2015; PTR-TOFMS, Stockwell et al., 2015). This paper synthesizes data four the four techniques to 1) compare the compositional space accessed and emission factors (EF) measured by the techniques, 2) provide comprehensive emission profiles for four fuels (Ponderosa Pine, Black Spruce, Indonesian Peat, and Chinese Rice Straw) and 3) characterize the volatility distribution of the emissions and identify potentially important secondary organic aerosol (SOA) precursors which have been understudied (e.g. in photochemical chamber experiments of SOA formation). This paper present new and very valuable findings for biomass burning and atmospheric chemistry fields. The study quantifies emissions for 10's of compounds that a have significant potential for SOA formation, but have received little or no attention in previous studies of SOA. By doing so this paper identifies potentially important gaps in our current understanding of BB impacts on SOA. The careful 4 instrument intercomparison presented in the paper may be useful for reinterpreting previous studies of BB that did not benefit from the unprecedented composition coverage and diversity of methods used here. I recommend this paper for publication pending consideration of the following comments.

Comments

R1.1: Please provide details on PP and BS fuels. Were they both boughs? How fresh? Was combustion limited to mostly needles and fine branch wood?

AC1.1: We have added notes on Page 3, lines 13-15 that pine and spruce boughs were burned during the selected fires and point the readers to our previous papers for more detailed fuel and fire descriptions.

R1.2: Pg 4 line 19. "chemical perturbation" does this include lamps? Was the chamber

dark during influx of emissions and prior to acquisition of WAS?

AC1.2: Yes, the chamber was dark prior to and during the WAS sample collection. We have clarified this point in the text on line Page 4, line 21: "WAS samples were collected from dark smog chambers."

R1.3: Figure 2b is messy and difficult to read. I suggest modifying to include only a handful of species that coincide with the main points of the corresponding text (P9, L24 – P10 L5)

AC1.3: We have replaced the text markers with shapes and O:C colorscale in Fig. 2b to improve the clarity.

R1.4: Any thought on why the high contribution of alkanes for peat versus other fuels?

AC1.4: The higher relative contribution of alkanes in peat smoke is possibly attributed to the fact that peat is composed of partially decayed plant matter. Indeed, peat is the first stage of coal formation and significant levels of n-alkanes have been measured in particulate matter derived from combustion of coal of various maturity levels (Oros and Simoneit, 2000). However, speciated measurements of additional peat fires should be performed to better understand the variability of alkane emissions from peat burning, and therefore we have not added this speculation to the current manuscript.

R1.5: P11, L19-33: Does the significant interference of C6H8 compounds at m/z 81 identified in this study suggest the previously reported MT EF using PTRTOFMS (Stockwell et al., 2015) should be revised? "The MT emission factors reported by Stockwell et al. (2015) were calibrated using m/z 81 due to the high degree of fragmentation of the $\alpha$-pinene standard under the PTRTOFMS drift tube conditions utilized during FLAME-4."

AC1.5: Although the PTR-TOFMS-derived monoterpene EF based on m/z 137 agreed significantly better with the GCxGC-TOFMS monoterpene EF, significant variability between the instruments was still evident, including potential interference at m/z 137 (e.g.,

from bornyl acetate, as discussed in the manuscript). Therefore, we primarily seek to emphasize the potential interferences in the PTR-TOFMS determination of monoterpene emissions from biomass burning.

R1.6: P14, L33: The author not the OA concentrations were 1000-6000ug/m3. Where do these numbers come from? Are these form previously published FLAME-4 papers? Please explain the origin of the OA measurements.

AC1.6: The OA concentrations are based on filter measurements performed by the University of Iowa, which is now noted on Line Page 15, line 12. Although the OA analyses have not been published, the sampling protocol has been described and the citation is now included in the manuscript.

References:

Oros, D. R., and Simoneit, B. R. T.: Identification and emission rates of molecular tracers in coal smoke particulate matter, Fuel, 79, 515-536, Doi 10.1016/S0016-2361(99)00153-2, 2000.
* * *

---

## Author Comment (AC2) · 2 Dec 2016

We thank the reviewer for the thoughtful comments. Their comments are reproduced here (R), with our responses noted (AC).

Anonymous Referee #2 The authors present emission factors for 4 different types of biomass based on laboratory measurements using 4 different methods to provide a greater coverage of organic gases in terms of mass and type. Methods are inter-compared and composite emission factors and speciation profiles are provided for the different biomass types. This is useful work for the fire science community and should

be published; however, several suggestions are made that would make this manuscript more broadly useful toward modeling wildland fire atmospheric impacts.

R2.1: Emission factors for biomass burning are known to vary by combustion component (e.g., flaming to smoldering). It would be useful for the authors to describe the combustion efficiency of the samples as a function of measured CO and CO2 or if that is not available as part of the experiment some qualitative description about the relative amount of time the biomass samples were flaming or smoldering. This will allow for a more specific application of these emission factors in modeling in the future as they may be more relevant for either the flaming or smoldering component of the various biomass types included as part of this assessment.

AC2.1: We have added the modified combustion efficiency value for each burn on Page 3, lines 19-22 and to Figure 7.

R2.2: From a modeling perspective, particle phase measurements would ideally be collected at the same time and at the same conditions as the gas phase measurements. Biomass burning clearly emits organic gases that are known to be a source of SOA in the atmosphere. However, the authors ignore ambient based field study data in their discussion of SOA formation potential that suggests little or no SOA is formed from biomass burning: Jolleys et al., 2012, ES&T, Forrister et al., 2015,GRL, Yu et al., 2016, JGR, Liu et al., 2016, JGR, and Cubison et al., 2011, ACP. This work is intended to provide emission factors and not be a chamber study looking at SOA formation so this comment is not meant to require significant changes, just to recognize that SOA formation from biomass burning in the atmosphere based on field study data is highly variable in terms of context. Perhaps IVOC emissions from biomass burning may not be that important for understanding SOA. However, future work should include measuring both gas (via the multi-instrument approach suggested here) and particle phases simultaneously to accounting for the elusive missing mass (capturing it regardless of phase).

AC2.2: Indeed, SOA formation observed in biomass burning plumes is highly variable, with some studies showing net loss of OA mass and others reporting large enhancements of OA in aged plumes. We have added this additional context on Page 2, lines 19-22: "Given that BB is the second largest source of NMOGs worldwide, the SOA formation potential from BB is large (Yokelson et al., 2008), yet observations of SOA formation in BB plumes have been highly variable, with OA mass enhancement ranging from <1 – 4 (Akagi et al., 2012; Forrister et al., 2015; Grieshop et al., 2009; Hennigan et al., 2011; Jolleys et al., 2012; May et al., 2015; Ortega et al., 2013; Vakkari et al., 2014; Yokelson et al., 2009)." We further note that such wide variability in the observed SOA formation in biomass burning plumes supports the need for improved speciation and assessment of potential SOA precursors so that we can better understand the processes contributing to these differences in OA production/loss. Additionally, recent modeling work has demonstrated that SOA may contribute substantially to OA mass in ambient plumes even when there is no observed net increase in aerosol mass following atmospheric aging by compensating for the evaporative losses caused by dilution (Bian et al., 2016).

Regarding IVOCs, our main objective in this work was to demonstrate that IVOCs are present in biomass burning smoke, that they are often undetected/untargeted in measurement campaigns, and to provide an initial indication of their composition. This should provide a first step toward determining any potential contribution to SOA by IVOCs in biomass burning plumes.

Specific comments: R2.3: At the OA mass loadings the measurements were made at (1000-6000 ug/m3, p. 14 C2 line 32), 50-85% of the lowest volatility IVOC they measure ($C^*=10^3$) would be in the particle phase (or 9-37.5% of $C^*=10^4$). While they didn't measure much mass at the lowest volatility, partitioning to the particle phase would have an impact on their results.

AC2.3: We agree that this is an important consideration for calculating EFs of compounds that may partition to the particle phase. The EFs of the gaseous organic

species as well as particulate organic carbon would be affected by such partitioning. We have added on Page 15, lines 23-25 that our lowest volatility IVOCs likely exist to some extent in the particle phase, and to the extent that is the case the reported EFs would be higher under lower OA mass loadings.

R2.4: Including the IVOC EF in Fig 7 for each fuel type would help provide context on how the magnitude and composition of IVOC emissions changes across the fuels.

AC2.4: We have added the total IVOC EFs under each pie chart of Fig. 7.

R2.5: In the instrument to instrument comparison, the authors note difficulty estimating monoterpene emission factors due to interference from other compounds and the need to account for fragmentation. The novel approach presented here would seem to include sesquiterpenes which are not specifically discussed even though these compounds have notable SOA yields. Did these approaches do well at capturing sesquiterpenes? Are there similar fragmentation issues related to appropriately characterizing the emissions of of sesquiterpenes? There are not a lot of C15 compounds for these 4 biomass types (only 3 seem to have C15) but perhaps the sesquiterpenes are captured here as fragments.

AC2.5: Up to 10 sesquiterpene isomers were observed by GCxGC-TOFMS in the both the pine and spruce smoke samples ($C_{15}H_{24}$ compounds listed in Table S1); the analysis of these compounds was discussed in detail in our previous publication (Hatch et al., 2015). The reviewer raises an interesting question regarding fragmentation of sesquiterpenes in PTR-TOFMS. The protonated molecular ion of sesquiterpenes at m/z 205 was not observed in these FLAME-4 PTR-TOFMS measurements. Demarcke et al. (2009) demonstrated that protonated sesquiterpene ions can fragment extensively at the electric field strength (E/N) used in the PTR-TOFMS measurements during FLAME-4 (136 Td), however the protonated molecular ion remained the major ion observed, with molecular ion yields ranging from 30-65% among four different sesquiterpene isomers. Therefore it is unclear whether the protonated molecular ion

was below the PTR-TOFMS detection limit during FLAME-4 or whether the molecular ion yields were significantly lower than indicated by Demarcke et al. (2009).

R2.6: In the SOA yields section, the authors note that furans are an important class of understudied organic gases in terms of SOA yield. In terms of this information being used for modeling atmospheric aerosol formation, is it possible that furans are chemically converted to traditional SOA precursors at time scales much faster than time steps in most operational air quality models and could simply be treated as a traditional SOA precursor?

AC2.6: We find no evidence that furans were converted to any known SOA precursors as smoke was stored in the dark combustion chamber over the 1-2 hour duration of the room burns included here. It is also unlikely that emitted furans could be chemically converted under oxidizing atmospheric conditions to traditional SOA precursors, which are mostly hydrocarbons (aromatics/alkanes/alkenes). However, furans react rapidly with OH radical and some of the reaction products are similar to those produced through oxidation of aromatic hydrocarbons (i.e., unsaturated dicarbonyls) (Bierbach et al., 1995; Bierbach et al., 1994). Though we note that the oxidation mechanism of furfural, the most abundant furan derivative, is currently unknown. So, it is certainly possible that in a chemical transport model the best way to represent the SOA formation would be to lump them with an appropriate existing SOA precursor(s), but ideally, additional information regarding SOA yields would be available to determine the best surrogate compounds.

References:

Akagi, S. K., Craven, J. S., Taylor, J. W., McMeeking, G. R., Yokelson, R. J., Burling, I. R., Urbanski, S. P., Wold, C. E., Seinfeld, J. H., Coe, H., Alvarado, M. J., and Weise, D. R.: Evolution of trace gases and particles emitted by a chaparral fire in California, Atmos. Chem. Phys., 12, 1397-1421, 2012.

Bian, Q., Jathar, S. H., Kodros, J. K., Barsanti, K. C., Hatch, L. E., May, A. A., Kreidenweis, S. M., and Pierce, J. R.: Secondary organic aerosol formation in biomass-burning plumes: Theoretical analysis of lab studies and ambient plumes, Atmos. Chem. Phys. Discuss., 2016, 1-36, 10.5194/acp-2016-949, 2016.

Bierbach, A., Barnes, I., Becker, K. H., and Wiesen, E.: Atmospheric chemistry of unsaturated carbonyls: butenedial, 4-oxo-2-pentenal, 3-hexene-2,5-dione, maleic anhydride, 3H-furan-2-one, and 5-methyl-3H-furan-2-one, Environ. Sci. Technol., 28, 715-729, 1994.

Bierbach, A., Barnes, I., and Becker, K. H.: Product and kinetic study of the OH-initiated gas-phase oxidation of furan, 2-methylfuran, and furanaldehydes at 300K, Atmos. Environ., 29, 2651-2660, 1995.

Demarcke, M., Amelynck, C., Schoon, N., Dhooghe, F., Van Langenhove, H., and Dewulf, J.: Laboratory studies in support of the detection of sesquiterpenes by proton-transfer-reaction-mass-spectrometry, Int. J. Mass. Spectrom., 279, 156-162, 10.1016/j.ijms.2008.10.023, 2009.

Forrister, H., Liu, J., Scheuer, E., Dibb, J., Ziemba, L., Thornhill, K. L., Anderson, B., Diskin, G., Perring, A. E., Schwarz, J. P., Campuzano-Jost, P., Day, D. A., Palm, B. B., Jimenez, J. L., Nenes, A., and Weber, R. J.: Evolution of brown carbon in wildfire plumes, Geophys. Res. Lett., 42, 4623-4630, 10.1002/2015gl063897, 2015.

Grieshop, A. P., Logue, J. M., Donahue, N. M., and Robinson, A. L.: Laboratory investigation of photochemical oxidation of organic aerosol from wood fires 1: measurement and simulation of organic aerosol evolution, Atmos. Chem. Phys., 9, 1263-1277, 2009.

Hatch, L. E., Luo, W., Pankow, J. F., Yokelson, R. J., Stockwell, C. E., and Barsanti, K. C.: Identification and quantification of gaseous organic compounds emitted from biomass burning using two-dimensional gas chromatography-time-of-flight mass spectrometry, Atmos. Chem. Phys., 15, 1865-1899, DOI 10.5194/acp-15-1865-2015, 2015.

Hennigan, C. J., Miracolo, M. A., Engelhart, G. J., May, A. A., Presto, A. A., Lee,

T., Sullivan, A. P., McMeeking, G. R., Coe, H., Wold, C. E., Hao, W. M., Gilman, J. B., Kuster, W. C., de Gouw, J., Schichtel, B. A., Collett, J. L., Kreidenweis, S. M., and Robinson, A. L.: Chemical and physical transformations of organic aerosol from the photo-oxidation of open biomass burning emissions in an environmental chamber, Atmos. Chem. Phys., 11, 7669-7686, DOI 10.5194/acp-11-7669-2011, 2011.

Jolleys, M. D., Coe, H., McFiggans, G., Capes, G., Allan, J. D., Crosier, J., Williams, P. I., Allen, G., Bower, K. N., Jimenez, J. L., Russell, L. M., Grutter, M., and Baumgardner, D.: Characterizing the aging of biomass burning organic aerosol by use of mixing ratios: A meta-analysis of four regions, Environ. Sci. Technol., 46, 13093-13102, 2012.

May, A. A., Lee, T., McMeeking, G. R., Akagi, S., Sullivan, A. P., Urbanski, S., Yokelson, R. J., and Kreidenweis, S. M.: Observations and analysis of organic aerosol evolution in some prescribed fire smoke plumes, Atmos. Chem. Phys., 15, 6323-6335, 10.5194/acp-15-6323-2015, 2015.

Ortega, A. M., Day, D. A., Cubison, M. J., Brune, W. H., Bon, D., de Gouw, J. A., and Jimenez, J. L.: Secondary organic aerosol formation and primary organic aerosol oxidation from biomass-burning smoke in a flow reactor during FLAME-3, Atmos. Chem. Phys., 13, 11551-11571, 2013.

Vakkari, V., Kerminen, V. M., Beukes, J. P., Tiitta, P., van Zyl, P. G., Josipovic, M., Venter, A. D., Jaars, K., Worsnop, D. R., Kulmala, M., and Laakso, L.: Rapid changes in biomass burning aerosols by atmospheric oxidation, Geophys. Res. Lett., 41, 2644-2651, 10.1002/2014gl059396, 2014.

Yokelson, R. J., Christian, T. J., Karl, T. G., and Guenther, A.: The tropical forest and fire emissions experiment: laboratory fire measurements and synthesis of campaign data, Atmos. Chem. Phys., 8, 3509-3527, 2008.

Yokelson, R. J., Crounse, J. D., DeCarlo, P. F., Karl, T., Urbanski, S., Atlas, E., Campos, T., Shinozuka, Y., Kapustin, V., Clarke, A. D., Weinheimer, A., Knapp, D. J., Montzka,

[Figure]

D. D., Holloway, J., Weibring, P., Flocke, F., Zheng, W., Toohey, D., Wennberg, P. O., Wiedinmyer, C., Mauldin, L., Fried, A., Richter, D., Walega, J., Jimenez, J. L., Adachi, K., Buseck, P. R., Hall, S. R., and Shetter, R.: Emissions from biomass burning in the Yucatan, Atmos. Chem. Phys., 9, 5785-5812, 2009.
* * *